# Millennial-scale denudation rates in the Himalaya of Far Western Nepal

Lujendra Ojha[1], Ken L. Ferrier[2], Tank Ojha[3]

[1]Department of Earth and Planetary Sciences, Johns Hopkins University, Baltimore, Maryland, USA.
[2]Department of Geoscience, University of Wisconsin, Madison, WI, USA.
[3]Department of Geoscience, University of Arizona, Tucson, AZ, USA.

*Correspondence to*: Lujendra Ojha (luju@jhu.edu)

**Abstract.** The Himalayas stretch ~3000 km along the Indo-Eurasian plate boundary. Along-strike variations in the fault geometry of the Main Himalayan Thrust (MHT) have given rise to significant variations in the topographic steepness, exhumation rate, and orographic precipitation along the Himalayan front. Over the past two decades, rates and patterns of Himalayan denudation have been documented through numerous cosmogenic nuclide measurements in central and eastern Nepal, Bhutan, and northern India. To date, however, few denudation rates have been measured in Far Western Nepal—a ~300-km-wide region near the center of the Himalayan arc—which presents a significant gap in our understanding of Himalayan denudation. Here we report new catchment-averaged millennial-scale denudation rates inferred from cosmogenic $^{10}$Be in fluvial quartz at seven sites in Far Western Nepal. The inferred denudation rates range from 385 ± 31 t km$^{-2}$ yr$^{-1}$ (0.15 ± 0.01 mm yr$^{-1}$) to 8737 ± 2908 t km$^{-2}$ yr$^{-1}$ (3.3 ± 1.1 mm yr$^{-1}$), and, in combination with our analyses of channel topography, are broadly consistent with previously published relationships between catchment-averaged denudation rates and normalized channel steepness across the Himalaya. These data show that the denudation rate patterns in the Far Western Nepal are consistent with those observed in the Central and Eastern Nepal. The denudation rate estimates from the Far Western Nepal show a weak correlation with catchment-averaged specific stream power, consistent with a Himalaya-wide compilation of previously published stream power values. Together, these observations are consistent with a dependence of denudation rate on both tectonic and climatic forcings, and represent a first step toward filling an important gap in denudation rate measurements in Far Western Nepal.

## 1 Introduction

The denudation of tectonically active mountain belts is controlled by feedbacks between tectonics, climate, and topography (e.g. *Willett*, 1999; *Hilley and Strecker*, 2004; *Whipple and Meade*, 2006; *Roe and Brandon*, 2011). In Earth's largest mountain belt, the Himalaya, some studies suggest that denudation is strongly controlled by climate (*Wobus et al.*, 2003; *Hodges et al.*, 2004; *Thiede et al.*, 2004; *Vannay et al.*, 2004; *Huntington et al.*, 2006; *Clift et al.*, 2008; *Gabet et al.*, 2008; *Hirschmiller et al.*, 2014; *Olen et al.*, 2015) while others suggest it is dominantly controlled by tectonic forcings (*Burbank et al.*, 2003; *Godard et al.*, 2014; *Scherler et al.*, 2014; *Olen et al.*, 2015), and yet others suggest that the relative strength of climatic and tectonic controls vary along strike with climate and crustal structure (*Harvey et al.*, 2015; *Olen et al.*, 2016; *van der Beek et al.*, 2016).

Comprehensively testing hypotheses about these feedbacks along the full length of the Himalayan arc (e.g., *Hodges*, 2000; *Beaumont et al.*, 2001; *Hodges et al.*, 2004; *Harvey et al.*, 2015) requires comprehensive coverage of denudation rate measurements along the range (e.g., *Whipple*, 2009; *Herman et al.*, 2010). However, at present, the spatial coverage of denudation rate measurements across the Himalaya is high in some regions and low in others (Figure 1). Basin-averaged denudation rates have been measured in many sites in central Nepal, Bhutan, and northern India (*Vance et al.*, 2003; *Wobus et al.*, 2005; *Finnegan et al.*, 2008; *Lupker et al.*, 2017; *Godard et al.*, 2012, 2014; *Lupker et al.*, 2012; *Munack et al.*, 2014; *Scherler et al.*, 2014b; *West et al.*, 2014; *Morell et al.*, 2015a, 2017, *Olen et al.*, 2015, 2016; *Portenga et al.*, 2015; *Dietsch et al.*, 2015; *Le Roux-Mallouf et al.*, 2015; *Abrahami et al.*, 2016; *Adams et al.*, 2016; *Kim et al.*, 2017; *Dingle et al.*, 2018). There

is, however, a significant gap in denudation rate measurements in Far Western Nepal, which spans a ~300-km region near the center of the Himalayan arc (Figure 1). To date, the only millennial-scale denudation rates that have been measured in Far Western Nepal are in the Karnali River basin and are based on cosmogenic nuclide measurements in samples collected near the range front (*Lupker et al.*, 2012). As far as we are aware, no cosmogenic nuclide-based denudation rates have been

measured in other basins in Far Western Nepal or in its interior. Basin-averaged erosion rates have also been measured from fluvial sediment fluxes over decadal timescales on the lower Karnali River (*Andermann et al.*, 2012), and a recent study has inferred apparent exhumation rates over million-year timescales at a series of points along the Karnali River (*van der Beek et al.*, 2016). No other erosion rate measurements, however, have been made in Far Western Nepal over any timescale. The limited number of measurements in Far Western Nepal is partly due to the well-known limited accessibility of the region. Van

der Beek et al. (2016), for example, noted that many of their thermochronometric sampling sites were only accessible on foot. This has hindered efforts to test hypotheses about feedbacks between climate, tectonics, and topography across this portion of the Himalaya.

Previous studies suggest that the relative strengths of the controls on denudation rate in Far Western Nepal may differ from those in central Nepal. In central Nepal, the presence of a single, major mid-crustal ramp in the Main Himalayan Thrust

(MHT) (e.g., *Schulte-Pelkum et al.*, 2005; *Bollinger et al.*, 2006; *Nábělek et al.*, 2009; *Elliott et al.*, 2016) has given rise to a steep topographic gradient with spatially focused exhumation and orographic precipitation (*van der Beek et al.*, 2016). In Far Western Nepal, by contrast, the topography rises more gradually and induces a less intense focusing of orographic precipitation, and has been hypothesized to be a reflection of two distinct mid-crustal ramps, each smaller than the one in central Nepal (*Harvey et al.*, 2015; *van der Beek et al.*, 2016). This is consistent with apatite fission-track thermochronometric

measurements that show that Myr-scale exhumation rates and specific stream power are significantly higher and more spatially focused in central Nepal than in Far Western Nepal (*van der Beek et al.*, 2016). To the extent that along-strike variations in uplift and orographic precipitation influence the spatial patterns and magnitudes of denudation rates, they may also induce along-strike variations in the feedbacks between climate, tectonics, and topography. In this study, we report new basin-averaged denudation rate measurements inferred from cosmogenic $^{10}$Be in stream sediment in Far Western Nepal to better

understand denudation rate patterns in this segment of the Himalaya. Our measurements show that denudation rates in these basins are consistent with those both east and west of Far Western Nepal, suggesting similar controls on denudation across this portion of the Himalayan arc over millennial timescales, and they highlight the regions that may be most useful to target for future denudation rate measurements.

## 2 Field area

Our study area in Far Western Nepal lies within the Himalayan orogenic belt, the result of ongoing convergence between the Indian and Eurasian plates since ~45-55 Ma (*Powell and Conaghan*, 1973; *Ni and Barazangi*, 1984; *Patriat and Achache*, 1984; *Coward and Butler*, 1985; *Mattauer*, 1986; *Rowley*, 1996; *Hodges*, 2000). This collision has resulted in >2500 km of crustal shortening (*Achache et al.*, 1984; *Besse et al.*, 1984; *Patriat and Achache*, 1984; *Besse and Courtillot*, 1988; *Clark*, 2012), with as much as >900 km of shortening accommodated across the Himalaya in western Nepal (*Lyon-Caen and*

*Molnar*, 1985; *Bilham et al.*, 1997; *Lavé and Avouac*, 2000; *DeCelles et al.*, 2001; *Robinson et al.*, 2006). Along the swath region shown in Figure 1, the topography of Far Western Nepal ranges from 1 to 7 km (Figure 2A). The convergence has led to a series of south-vergent thrust faults that define the boundaries of four major tectonostratigraphic units that encompass the study basins (*Heim and Gansser*, 1939; *Gansser*, 1964): (i) the Subhimalaya, (ii) Lesser Himalaya, (iii) Greater Himalaya, and (iv) Tibetan Himalaya (Figure 3B).

These tectonostratigraphic units are characterized by distinctive topographic, lithologic, and climatic characteristics. The Subhimalaya is at the southern edge of the Himalaya, bounded by the Main Frontal thrust (MFT) at the Himalayan topographic front to the south and the Main Boundary thrust (MBT) to the north. Its bedrock is dominated by the Siwalik Group, a sequence of north-dipping thrust sheets of Neogene fluvial and alluvial deposits dated to 12.5-2.8 Ma (*Ojha et al.*, 2009). Its topography is relatively subdued, with the lowest mean elevation, hillslope gradient, relief, and normalized channel

steepness index $k_{sn}$ (Equation 1) among the four Himalayan tectonostratigraphic units in western Nepal (*Harvey et al.*, 2015).

$$S = k_{sn} A^{-\theta_{\text{ref}}} \tag{1}$$

Here $k_{sn}$ is defined as the proportionality constant between channel gradient $S$ and drainage area $A$ to the power $-\theta_{\mathrm{ref}}$, which is known as the reference concavity index (*Wobus et al.*, 2006; *DiBiase et al.*, 2010) and often taken to be a commonly adopted reference value of 0.45 (e.g., *Scherler et al.*, 2017).

North of the Subhimalaya is the Lesser Himalaya, which lies between the MBT to the south and the Main Central Thrust (MCT) to the north. Its bedrock is Proterozoic in age (*DeCelles et al.*, 2000) and comprises three major stratigraphic units: i) the Lesser Himalayan sequence of Paleoproterozoic-Neoproterozoic siliciclastic and carbonate strata; ii) the Gondwana sequence of Permian-Paleocene siliciclastic strata; and iii) the foreland basin sequence of Eocene-lower Miocene siliciclastic strata. In western Nepal, the Lesser Himalayan sequence is interrupted by exposed bedrock of the Greater Himalayan and Tibetan Himalayan sequences (*Gansser*, 1964; *DeCelles et al.*, 2001; *Robinson et al.*, 2006). Mean elevations in the Lesser Himalaya in western Nepal are ~1-2 km south of this exposure and ~2-3 km north of this exposure. The topography of the lesser Himalaya is characterized by higher mean hillslope gradients, relief, and normalized channel steepness than the Subhimalaya, though not as high as those in the Greater Himalaya (*Harvey et al.*, 2015).

The Greater Himalaya lies generally northeast of the MCT and southwest of the South Tibetan Detachment (STD), aside from its appearance in the Lesser Himalaya in the Dadeldhura Klippe. It is primarily composed of upper amphibolite-grade metasedimentary and meta-igneous rocks dominated by pelitic gneiss, marble, calc-silicate rocks, and granitic orthogneisses (*Pecher*, 1989; *Vannay and Hodges*, 1996). In central Nepal, the boundary between the Lesser and Greater Himalaya is marked by a physiographic transition known as PT2, where elevations rise sharply from ~2-3 km to >8 km over a distance of 40-50 km (*Hodges et al.*, 2001, 2004). In Far Western Nepal, by contrast, the topography is characterized by two physiographic transitions rather than one. These have been termed PT2-S to the south, where mean elevations increase from ~1.5 km to ~3 km over a distance of ~20-30 km, and PT2-N roughly 75 km north of PT2-S, where mean elevations increase from ~3 km to ~5 km (*Harvey et al.*, 2015). Mean hillslope gradients, relief, and normalized channel steepness are generally higher in the Greater Himalaya than in the other tectonostratigraphic provinces (Figures 3-4) (*Harvey et al.*, 2015).

The northernmost tectonostratigraphic unit in western Nepal is the Tibetan Himalaya, which lies north of the STD. Its bedrock is dominated by low-grade to unmetamorphosed rocks including marble, dolostone, shale, and sandstones (*Robinson et al.*, 2006). Its topography is characterized by high elevations (~5 km) but lower mean relief, hillslope gradient, and normalized channel steepness than both the Greater and Lesser Himalaya (*Harvey et al.*, 2015).

Precipitation rates and patterns in the Himalaya are dominantly orographically controlled, with moisture-laden winds rising up the range from the south and dropping out most of their moisture before reaching the Tibetan Himalaya to the north (e.g., *Bookhagen and Burbank*, 2010). Within our study basins in western Nepal, mean annual rainfall (MAR) inferred from TRMM 2B31 remote sensing data is highest in the Subhimalaya with a spatial average of ~2.6 m (*Bookhagen and Burbank*, 2010). The Lesser Himalaya and Greater Himalaya are relatively drier, with spatially averaged MAR of ~1.8 m and ~0.6 m, respectively, while the Tibetan Himalaya lies beyond the peak in orographic rainfall and is significantly drier, with a spatially averaged MAR of ~0.3 m (*Bookhagen and Burbank*, 2010). Modern glaciers in western Nepal are found only at elevations greater than ~2800 m, and thus are largely restricted to the Tibetan and Greater Himalaya (GLIMS and National Snow and Ice Data Center, 2005; Figure S7).

## 3 Methods

### 3.1. Sample collection, preparation, and analysis

We collected stream sediment from active fluvial sediment bars in February and March 2015 at seven locations in Far Western Nepal (Table 1; Figure 3), sampling sediment with a grain size distribution typical of the visible sediment on each bar (Table S1). The basins upstream of our sample sites range in size from ~237 to ~24,565 km$^2$, some of which lie entirely within the Lesser Himalaya (Budhiganga, Kalanga), while others span multiple tectonostratigraphic units, including the largest (Karnali), which spans the Subhimalaya to the Tibetan Himalaya (Figures 3, S1). From these samples, we extracted 59-154 g of quartz from the >250-micron size fraction through standard magnetic and chemical separations (*Kohl and Nishiizumi*, 1992). The samples were dominated by sand-size sediment (Table S1), and we analyzed quartz in all sediment grain sizes to avoid introducing biases that would be associated with analyzing only a single grain size (e.g., *Brown et al.*, 1995; *Riebe et al.*, 2015). After verifying the purity of the quartz samples by measuring Al concentrations by ICP-OES, we spiked quartz samples with 247-262 micrograms of $^9$Be in a 3 N HNO$_3$ low-$^{10}$Be background carrier (GFZ standard 2Q2P14/10/2010, Hella Wittmann-Oelze, personal communication, October 2015), and dissolved the mixture in trace metal grade hydrofluoric and nitric acids.

After dissolution, we added trace metal grade sulfuric acid to the solution, dried it down in platinum crucibles, and redissolved it in hydrochloric acid. Beryllium was isolated from other elements through pH adjustment with NaOH and oxalic acid, centrifugation, and cation exchange column chemistry. We converted BeOH to BeO by baking the samples in quartz crucibles, mixed the BeO with niobium powder, and packed the mixture in targets at the University of Wyoming Cosmogenic Nuclide sample preparation facility. $^{10}Be/^9Be$ ratios in the targets were measured at PRIME lab at Purdue University in August 2017 (Table 2) and corrected for a process blank ($^{10}Be/^9Be$ ratio of $(6.338 \pm 1.269) \times 10^{-15}$).

## 3.2. Computation of denudation rates from measured $^{10}Be$ concentrations

Basin-averaged denudation rates are commonly inferred from $^{10}Be$ concentrations in detrital quartz under the assumption of steady denudation (*Lal*, 1991; *Brown et al.*, 1995; *Bierman and Steig*, 1996; *Granger et al.*, 1996). Here we use the same approach to estimate steady-state denudation rates with the CRONUS v2.3 calculator (*Balco et al.*, 2008), a community-standard tool for computing steady denudation rates from cosmogenic nuclide concentrations at a given latitude and altitude. When applied to basin-averaged denudation rates, the CRONUS v2.3 calculator requires as input a number of characteristics of the sampled basin: the mean basin latitude and longitude, thickness of the sample, bedrock density, mean basin shielding factor, and the effective basin elevation, which is the elevation at which the $^{10}Be$ production rate at the mean basin latitude equals the basin-averaged $^{10}Be$ production rate (*Balco et al.*, 2013). This requires computing the production rate at every point within the basin, which in turn requires computing the shielding factor at every point within the basin. Here we describe how we quantified shielding due to topography, glaciers, and seasonal snow. In the Discussion section, we describe how our denudation rate estimates may be affected by uncertainties in a variety of factors, including potential lithologic variations in quartz abundance, which are not well quantified across our study basins.

### 3.2.1. Shielding by glacier ice

The GLIMS database (*GLIMS and National Snow and Ice Data Center*, 2005) shows that glaciers cover small portions of the sampled basins at present, accounting for an average of less than 4% of the sampled basins' drainage areas (range 0-6.9%; Table 1; Figure S7). Following previous studies in the Himalaya (e.g., *Lupker et al.*, 2012), we compute denudation rates under the approximation that glaciers completely shield the underlying rock from cosmogenic radiation, such that sediment generated by subglacial erosion has a $^{10}Be$ concentration of 0 atoms $g^{-1}$. This approximation is justified by the large thickness of mountain glaciers (average thickness of > tens of meters; e.g., *Sanders et al.*, 2010) relative to the characteristic penetration depth of cosmogenic neutrons through ice. For a glacier with a thickness of 50 m, for example, $^{10}Be$ production by neutron spallation under the ice should be approximately $10^{-12}$ times that at the surface.

The fraction of subglacially derived quartz in our samples is unknown. In the absence of evidence to suggest otherwise, we assume that the sampled sediment contains subglacial sediment in proportion to the fraction of the basin under ice, which we denote as $f_{ice}$. This is equivalent to assuming that the sediment delivery rate to the channel network in the ice-covered regions is the same as that in the ice-free regions. This assumption may overestimate the amount of glacially-derived sediment relative to the areal proportion of glaciated terrain (e.g., due to grain size fining during erosion and transport; *Olen et al.*, 2015) or underestimate it (e.g., due to faster erosion under glaciers; *Godard et al.*, 2012).

Under these assumptions, all $^{10}Be$ in our samples should be derived from the ice-free portions of the basins. We therefore infer denudation rates using estimates of $^{10}Be$ concentrations in quartz derived from the ice-free portions of the basins. Concentrations of $^{10}Be$ in the ice-free regions should be slightly higher than the measured concentrations due to dilution of $^{10}Be$ in the measured sample by the incorporation of subglacially derived quartz. We compute $^{10}Be$ concentrations in the ice-free regions as $N_{ice\text{-}free} = N(1 - f_{ice})^{-1}$, where $N$ is the measured concentration. This yields $^{10}Be$ concentrations in the ice-free region that are 0-7% higher than the measured $^{10}Be$ concentrations (Table 2).

### 3.2.2. Shielding by topography and seasonal snow

In the ice-free portions of the sampled basins, $^{10}Be$ production rates may be reduced through shielding by topography and seasonal snow. We computed topographic shielding at each grid point within each basin by standard methods, using the algorithm for computing topographic shielding in the CRONUS v2.3 calculator with an azimuthal spacing of 10° (*Balco et al.*, 2008). As a basin-averaged quantity, topographic shielding factors range from 0.976 to 0.994 (Table 2), implying that topographic shielding reduces basin-averaged $^{10}Be$ production rates in the study basins by 0.6 – 2.4%. These values are consistent with topographic shielding factors in other Himalayan catchments (e.g., *Olen et al.*, 2015). Recently, DiBiase (2018)

showed that this approach can overestimate the extent of topographic shielding, particularly in steeply dipping catchments, and argued that topographic shielding factors should be 1 in basins with horizontal surrounding ridges. If this horizontal ridge geometry is applicable to our study basins, where our estimates of topographic shielding range from 0.9759 to 0.9939 (Table 2), then the denudation rates in Table 2 would be underestimates by 0.6% to 2.5%.

5         The corrections for shielding by seasonal snow may be significantly larger than the percent-level corrections for topographic shielding. The extent of shielding by snow, however, is not well constrained by field measurements, as previous studies in the Himalaya have shown (*Scherler et al.*, 2014). Estimating the effects of shielding by seasonal snow requires measurements of the area covered by snow as well as snow thickness and density (*Schildgen et al.*, 2005). As far as we are aware, these measurements do not exist in our study region. Scherler et al. (2014), noting the same absence of measurements

in their study region in the Garhwal Himalaya, used MODIS satellite observations to estimate plausible ranges of snow shielding factors in their study area. Here we follow the approach of *Scherler et al.* (2014) to estimate shielding by seasonal snow in our study basins, while stressing that significant uncertainties exist in the resulting estimates.

        We estimate snow shielding by applying two correlations measured by *Scherler et al.* (2014). The first is a nearly linear dependence of the annual duration of snow coverage $t_d$ (days per year) on elevation $z$ (m) (Figure S1b in *Scherler et al.,*

2014). This is well fit by the third-order polynomial $t_d = -9.3218 \cdot 10^{-9} z^3 + 1.0302 \cdot 10^{-4} z^2 - 0.2422 z + 172.8959$ ($R^2 = 0.9994$) in the elevation range considered by *Scherler et al.* (2014) (1.55 km to 4.84 km). This polynomial yields snow-cover durations of 10 days and 356 days at the lower and upper limits of this altitude range, respectively. We extend this empirical regression by applying fixed lower and upper limits of 0 days and 365 days at elevations < 1500 m and > 4920 m, respectively.

        The second correlation we apply is a negative dependence of the snow shielding factor $S_s$ on $t_d$ (Figure S2b in *Scherler*

*et al., 2014*), which is well fit by the sixth-order polynomial $S_s = -3.5198 \cdot 10^{-15} t_d^6 + 3.7235 \cdot 10^{-12} t_d^5 - 1.3853 \cdot 10^{-9} t_d^4 + 2.1045 \cdot 10^{-7} t_d^3 - 1.4445 \cdot 10^{-5} t_d^2 + 3.0228 \cdot 10^{-4} t_d + 0.9979$ ($R^2 = 0.9998$). We apply these two polynomials to each grid point in the study basins to compute a grid of snow shielding factors for each basin.

        We multiply the resulting snow shielding grid to the topographic shielding factor grid to compute grids of the total shielding factor in the ice-free regions of the basins. From this product, we compute the basin-averaged value of the total

shielding factor. These basin-averaged values range from 0.777 to 0.992 (Table 2), implying that seasonal snow can reduce [10]Be production rates by as little as ~1% to as much as >20% in the study basins. Like *Scherler et al.* (2014), we stress that these calculations are largely unconstrained by field measurements of snow thickness and density and that these estimates of snow shielding are mainly used to show that snow shielding can be significantly larger than topographic shielding. We neglect shielding by vegetation because vegetative shielding is negligible except in forests with exceptionally high vegetation density

by mass (*Ferrier et al.*, 2005).

### 3.2.3. Computation of denudation rates

        Using the shielding factor grids for the ice-free portions of the sampled basins, we computed the [10]Be production rate at each grid point within the basin – including shielding by topography and snow – with the CRONUS v2.3 calculator (*Balco*

*et al.*, 2008) and applying the Stone (2000)/Lal (1991) time-invariant scaling for latitude and altitude (*Lal*, 1991; *Stone*, 2000). We then computed the basin-average [10]Be production rate and used the CRONUS v2.3 calculator to determine the basin effective elevation, which is the elevation at which the [10]Be production rate under the basin-averaged shielding factor equals the basin-averaged production rate (*Balco et al.*, 2013). From the basin effective elevation, basin-average shielding factor, and basin-average production rate, we computed the basin-averaged denudation rate with the CRONUS v2.3 calculator at the basin-

average latitude and longitude. The denudation rate uncertainties in Table 2 are the "external" uncertainties reported by the CRONUS calculator for the Lal/Stone production rate scaling scheme, and include propagated uncertainties on the [10]Be production rate.

### 3.3. Topographic analyses

45         We computed several topographic characteristics of the study basins from the Shuttle Radar Topography Mission (SRTM) topographic dataset that spans the Himalaya with a horizontal resolution of ~30 meters. First, we computed the normalized channel steepness $k_{sn}$ (Equation 1), adopting a reference concavity $\theta_{ref}$ of 0.45, following the Himalaya-wide compilation of *Scherler et al.* (2017). In the context of the stream power law ($dz/dt = U - KA^m S^n$) and under spatially uniform rock uplift rate $U$ and erodibility $K$, at topographic steady state ($dz/dt = 0$) the normalized channel steepness can be interpreted

as $k_{sn} = (U/K)^{1/n}$. Here we use the methodology of *Perron and Royden* (2013) to estimate $(U/K)^{1/n}$ from the integral measure

$\chi$, following recent studies of channel steepness elsewhere in the Himalaya (*Olen et al.*, 2015; *Morell et al.*, 2017). We computed $\chi$ using Matlab algorithms developed by *Gallen and Wegmann* (2017) implemented in TopoToolbox2 (*Schwanghart and Scherler*, 2014), as this approach measures $k_{sn}$ with smaller uncertainty. For each sampled basin, we computed $k_{sn}$ for portions of the channel network that are likely to be bedrock-dominated channels (and hence most appropriate for the $\chi$ methodology, which was developed for bedrock-dominated channels subject to the stream power law) by restricting the analysis to drainage areas >1 km$^2$. Basin-averaged channel steepness was calculated as the mean of all $k_{sn}$ estimates for all channels and their tributaries within a basin. We used the same topographic data to compute topographic relief within a circular window of 5-km radius and basin-averaged hillslope gradients.

*3.4. Climate analyses*

We used processed Tropical Rainfall Measuring Mission (TRMM) 2B31 data calibrated to rain- and river-gauging data throughout the Himalayas (*Bookhagen and Burbank*, 2010) to compute basin-averaged precipitation rates at our study sites (Figure 3C). The TRMM 2B31 data is a record of mean annual precipitation from January 1998 to December 2005 at a spatial resolution of ~5 x 5 km and a temporal resolution of 1-3 times per day (*Bookhagen and Burbank*, 2010). We used the TRMM 2B31 data (Bookhagen and Burbank, 2010) and 90-m SRTM topographic data (Farr et al., 2007) to compute specific stream power $\omega = \rho g Q S w^{-1}$ along the channel network in each basin, where $\rho$ is the flow density, g is gravitational acceleration, $Q$ is river discharge, $S$ is channel gradient, and $w$ is channel width. Following Olen et al. (2016), we computed channel gradient as an 11-point mean along the channel, and we adopt the width-discharge relation $w = 3.4Q^{0.40}$ in Craddock et al. (2007), where $w$ has units of m and $Q$ has units of m$^3$ s$^{-1}$. We computed basin-average specific stream power as the mean of all specific stream power estimates within each basin's channel network at drainage areas >1 km$^2$.

*3.5. Landslide scar mapping*

To quantify the abundance of landslides in the study basins, landslide scars were manually mapped in high-resolution satellite imagery provided by Google Earth's historical imagery database from 2002-2015 (e.g., Figure S8). These scars were used to estimate each basin's fractional coverage by landslide scars, $f_{scar}$ (Table 1), calculated as the total area of landslide scars across all 2002-2015 satellite images divided by the basin area. All mapping was done on images that preceded the April 25, 2015 Gorkha earthquake, such that estimates of $f_{scar}$ are unaffected by landslides triggered by this earthquake. This mapping is part of a Nepal-wide landslide database currently in preparation.

**4. Results**

To illustrate the degree to which shielding by seasonal snow can affect our denudation rate estimates, we report two denudation rates for each basin (Table 2), one in which snow shielding is accounted for as in Section 3.2.2, and one in which it is neglected, as is common in other Himalayan erosion studies (e.g., *Lupker et al.,* 2012). Mean denudation rates inferred from the measured [10]Be concentrations range from 385 to 8737 t km$^{-2}$ yr$^{-1}$ when accounting for snow shielding, and from 385 to 9382 t km$^{-2}$ yr$^{-1}$ when neglecting snow shielding. The difference between mean denudation rate estimates with and without snow shielding ranges from 0 to 28% in any one basin. From this point forward, we restrict our discussion to the denudation rate estimates that include snow shielding, since we believe that these are likely more representative of the true shielding history and hence the true denudation rates.

We determined the normalized channel steepness indices $k_{sn}$ for all channels at drainage areas of >1 km$^2$ using the integral method on 30-m SRTM topographic data (Section 3.3). Basin-average $k_{sn}$ range from 127 to 217 m$^{0.9}$ among the study basins (Table 1). Maps of $k_{sn}$ within each basin are shown in Figure S2.

Catchment-averaged specific stream power values range from 48 W m$^{-2}$ in the Karnali basin to 145 W m$^{-2}$ in the Budhiganga basin (Figure 4C). Spatial variations in specific stream power are relatively large in the largest basins (Karnali, Karnali Ferry, Bheri), which span the relatively dry and low-gradient Tibetan Himalaya (where specific stream power is low; Figure 4C) as well as the relatively wet and high-gradient Lesser Himalaya, where specific stream power is relatively high. Spatial variations in stream power are smallest in the smallest basin (Raduwa), which lies entirely in a lower-gradient region of the Lesser Himalaya (Figure 3B).

Microseismicity is dominantly focused in a wide belt across the Karnali, Budhiganga, Kalanga, and Seti basins (Figure 3D). The frequency of microseismicity is particularly low in the Raduwa basin and across much of the Bheri basin (Table 1;

Figure S5), consistent with relatively weak tectonic forcing in the lowest-elevation portion of the Lesser Himalaya in the southwestern portion of Far Western Nepal (where the Raduwa basin lies) and in the eastern portion of Far Western Nepal (where the Bheri basin lies).

Landslide scars are present in each of the study basins and are clustered most densely in the southern portions of the Seti and Kalanga basins, as well as in the eastern portion of the Raduwa basin (Figure 4D). Inspection of the satellite imagery (Figure S6, S8) and our observations during sample collection in the field did not reveal landslide scars immediately adjacent to our sampling sites, consistent with sampled sediment that is not dominated by recent landslide-derived sediment in the study basins.

**5. Discussion: Potential sources of uncertainty in denudation rate estimates**

The conventional approach for estimating basin-average denudation rates from [10]Be in detrital quartz rests on two assumptions: that quartz grains approach the surface at a steady rate during exhumation (the steady erosion assumption; *Lal*, 1991), and that the sampled sediment is a well-mixed representation of all regions within the upstream basin (the well-mixed assumption; *Brown et al.*, 1995; *Bierman and Steig*, 1996; *Granger et al.*, 1996). Here we discuss how uncertainties in our denudation rate estimates may be affected by deviations from steady erosion and well-mixed conditions at our study sites, as well as by uncertainties in shielding by snow and ice.

*5.1. Uncertainties due to landsliding*

In rapidly eroding terrain, the steady erosion assumption can be violated by landsliding (*Brown et al.*, 1995; *Niemi et al.*, 2005; *Yanites et al.*, 2009; *West et al.*, 2014), which perturb steady [10]Be concentrations in fluvial sediment by supplying to rivers a mixture of low-[10]Be sediment from depth and high-[10]Be sediment from the near surface. These perturbations can be large. West et al. (2014), for example, documented a two- to threefold drop in [10]Be concentrations in fluvial quartz after the 2008 $M_w$ 7.9 Wenchuan earthquake, which triggered over 57,000 landslides (*Li et al.*, 2014). These observations are consistent with numerical studies showing that stream sediment should have [10]Be concentrations that are lower than the long-term average shortly after landsliding (*Niemi et al.*, 2005; *Yanites et al.*, 2009). If interpreted within the steady erosion framework, sediment that is rich in landslide-derived material would yield an inferred denudation rate higher than the true long-term average denudation rate.

An important corollary of this behavior is that if landsliding produces periods of time where [10]Be concentrations in stream sediment are lower than the long-term average, then there must be other times at which they are higher than the long-term average, simply by definition of the long-term average (*Niemi et al.*, 2005; *Yanites et al.*, 2009). Sediment collected during these intervals between landsliding would yield denudation rates that are lower than the long-term average. Thus, sediment collected from basins subject to landsliding may yield inferred denudation rates that underestimate or overestimate the true long-term average denudation rate, depending on the timing of sample collection relative to influxes of landslide material to the stream sediment.

During sample collection, we took care to avoid sampling sediment directly downstream of landslide scars or from deposits that appeared to be a result of landsliding, and our landslide mapping (Figure S6) does not present evidence that landslide-derived material should be overrepresented in our samples. The study basins have drainage areas of 237 – 24,565 km$^2$, significantly larger than the ~100 km$^2$ drainage area below which [10]Be concentrations are most likely to be strongly perturbed by landslide-derived sediment (*Niemi et al.*, 2005; *Yanites et al.*, 2009). Thus, in the absence of evidence supporting alternative interpretations of the measured [10]Be concentrations, we take the inferred denudation rates in Table 2 to be the most parsimonious interpretation of the measured [10]Be concentrations.

*5.2. Uncertainties due to chemical erosion*

A second process that can violate the steady erosion assumption is chemical erosion, which affects cosmogenic-based estimates of denudation rates in two ways. If chemical erosion happens at depths below the upper few meters where cosmogenic nuclides are dominantly produced, it will not be accounted for in measured [10]Be concentrations (*Riebe et al.*, 2003; *Dixon et al.*, 2009; *Ferrier et al.*, 2010). If chemical erosion happens preferentially to non-quartz minerals in the soil, it will increase the exposure time of quartz to cosmogenic radiation relative to other mineral phases, and will yield [10]Be concentrations in quartz that are higher than they would be if all minerals phases were dissolved at equal rates (*Small et al.*, 1999; *Riebe et al.*, 2001). These two effects have been combined into a single correction factor termed the Chemical Erosion Factor, or CEF

(*Riebe and Granger*, 2013), which quantifies the degree to which [10]Be-based denudation rates that neglect these effects would be in error.

We cannot measure the CEF at our study sites directly because we lack the measurements of immobile element concentrations in saprolite and its parent bedrock that would be needed to do so. Riebe and Granger (2013), however, documented a positive correlation between CEF and mean annual precipitation (MAP) within their global compilation of CEF measurements. Similarly, modern fluvial sediment and solute fluxes elsewhere in the Himalaya suggest that the chemical weathering flux in the Ganges and Brahmaputra Rivers is ~9 ± 2% of the suspended sediment flux (*Galy and France-Lanord,* 2001) and that chemical weathering fluxes in Himalayan basins may be small relative to those generated in the lowland floodplains (*West et al., 2002; Lupker et al., 2012*). To the extent that these measurements are applicable to our study basins, this suggests that chemical erosion may have only a small effect on our denudation rate estimates. Future measurements of immobile element concentrations in bedrock and saprolite at the study sites will be needed to verify these estimates.

*5.3. Uncertainties due to shielding by snow and ice*

An additional source of uncertainty is in the shielding of cosmogenic radiation by snow and ice. As described in Section 3.2.2, accounting for shielding by seasonal snow can significantly affect the inferred denudation rates in some of our sampled basins, with estimated denudation rates as much as ~30% lower in some basins after accounting for seasonal snow. These estimates of snow shielding, however, are largely unconstrained by local measurements of snow coverage, thickness, and density, which highlights the need for measurements of snow cover, thickness, and density in this region.

We are likewise unaware of estimates of paleo-glacier extent in the study basins, which would have affected the extent of shielding by ice. The relatively rapid erosion in the study basins, however, imply that modern glacier extents may provide a reasonable approximation for ice shielding in our samples. The characteristic timescale of [10]Be accumulation is the characteristic penetration depth of cosmogenic neutrons (160 g cm$^{-2}$; Gosse and Phillips, 2001) divided by the denudation rate (e.g., Granger et al., 1996). Our denudation rate estimates imply characteristic denudation timescales that range from 183 ± 61 years to 4156 ± 452 years, implying that that glacier coverage during the last glacial period is unlikely to be relevant for our samples. This suggests that our estimates of ice shielding based on the modern glacier coverage may be reasonable approximations of the ice shielding conditions over the past few hundreds to thousands of years.

*5.4. Uncertainties due to spatial variations in lithology*

The well-mixed assumption can be violated by the existence of spatially variable lithologies in a basin, since they imply that quartz in stream sediment may not be sourced uniformly within the basin (*Bierman and Steig*, 1996). We are unaware of published quartz abundances in the underlying lithologies, so we do not recalculate denudation rate estimates for our study basins here, and note that this provides motivation for further field mapping in western Nepal. Instead, we illustrate how large this effect might be by considering two hypothetical end-member scenarios. Among our study sites, the basin that is most likely to be affected by spatially variable lithologies is the Kalanga basin, which is underlain by quartz-bearing lithologies at high elevations to the north and low elevations to the south, but is dominated by quartz-poor carbonate and slate/shale lithologies in between (Figure S1; Robinson et al., 2006). Here we show how the inferred denudation rates for Kalanga would differ if the quartz had been sourced only from the quartz-bearing lithologies at high or low elevation.

As with our other analyses, we compute denudation rates for these hypothetical scenarios using the CRONUS v2.3 calculator (*Balco et al.*, 2008). If the sampled quartz had been sourced only from the high-elevation lithologies, the effective basin elevation would be 4287 m (higher than the total-basin effective elevation of 2742 m; Table 2), $f_{ice}$ would be 34.4% (higher than the total-basin estimate of 1.7%), and the total shielding factor for the ice-free region would be 0.733 (lower than the total-basin estimate of 0.930). Similarly, if the sampled quartz had been sourced only from the low-elevation lithologies, the effective basin elevation would be 2123 m, $f_{ice}$ would be 0%, and the total shielding factor would be 0.979. These parameters would yield inferred denudation rates of 2151 ± 266 t km$^{-2}$ yr$^{-1}$ and 1370 ± 167 t km$^{-2}$ yr$^{-1}$ for the high-elevation and low-elevation scenarios, respectively, or 1.17 ± 0.20 times and 0.75 ± 0.13 times that of the full-basin estimate of 1832 ± 224 t km$^{-2}$ yr$^{-1}$, respectively (Table 2). Even in these extreme scenarios, the inferred denudation rate would only be in error by no more than ~25%, suggesting that spatial variations in quartz abundance are unlikely to be a major source of error in the inferred denudation rates. In this scenario, the inferred ksn and specific stream power estimates would be 46% and 44% higher in the high-elevation portion of the Kalanga basin, respectively, and 33% and 39% lower in the low-elevation portion of the Kalanga

basin, respectively. Like the differences in the inferred denudation rates, these differences are relatively small compared to the scatter in the trends in Figure 5, suggesting that these effects are unlikely to be a major source of scatter in Figure 5.

*5.5. Measurements in Karnali are broadly consistent with well-mixed sediment*

Our measurements provide some limited evidence in support of the well-mixed assumption. Two of our samples, Karnali and Karnali Ferry, are taken from the main stem of the Karnali River. The Karnali Ferry sample was collected ~30 km upstream of the Karnali sample, such that the 24,052 km$^2$ basin upstream of the Karnali Ferry sample is contained entirely within the 24,565 km$^2$ basin upstream of the Karnali sample (Figure 3). Since the sampled basins overlap with one another and are nearly the same size, the denudation rates inferred from these samples pertain to nearly the same source areas, and
therefore provide an opportunity to document the natural variability in denudation rates within a large basin. After accounting for shielding by snow, our measurements yield denudation rates of 2044 ± 209 t km$^{-2}$ yr$^{-1}$ in the Karnali basin and 1717 ± 157 t km$^{-2}$ yr$^{-1}$ in the Karnali Ferry basin. This agreement within uncertainty is consistent with the well-mixed assumption, at least along the ~30-km reach between the Karnali and Karnali Ferry sites.

        The well-mixed assumption can be further assessed by comparison of our Karnali sample with a sample collected by
Lupker et al. (2012) <1 km upstream of Karnali (sample CA10-5 in Lupker et al., 2012). Lupker et al. (2012) measured a $^{10}$Be concentration of 48,300 ± 8700 atoms g$^{-1}$ at this site, a factor of 1.78 ± 0.34 times higher than our measured concentration of 27,127 ± 1702 atoms g$^{-1}$ at Karnali. The difference between our measured $^{10}$Be concentration and that in Lupker et al. (2012) is consistent with the temporal variations in stream sediment $^{10}$Be concentrations noted by Lupker et al. (2012). In addition, it is possible that grain size differences between samples may have contributed to the difference in $^{10}$Be concentrations, as Lupker
et al. (2012) analyzed quartz grains in the 125-250-micron size fraction, whereas we analyzed only grains larger than 250 microns. This would be consistent with the often-observed negative correlations between $^{10}$Be concentrations and grain size (*Brown et al.*, 1995; *Belmont et al.*, 2007; *Puchol et al.*, 2014; *Riebe et al.*, 2015; *Lukens et al.*, 2016). Even if there were no negative correlation between $^{10}$Be concentrations and grain size in Karnali sediment, however, this difference is relatively small compared to the order of magnitude differences frequently observed between millennial-scale and decadal-scale erosion
rates (*Kirchner et al.*, 2001; *Hewawasam et al.*, 2003; *Covault et al.*, 2013; *Marc et al.*, 2019). In summary, we suggest that this difference may be a reflection of the natural variability in $^{10}$Be concentrations in fluvial sediment, while noting that it is also consistent with a grain-size dependency of $^{10}$Be in fluvial quartz.

**6. Comparison of inferred denudation rates to previously published rates**

The $^{10}$Be-derived denudation rates at our study sites (385 ± 31 to 8737 ± 2908 t km$^{-2}$ yr$^{-1}$) are within the range of published Himalayan denudation rates in central and eastern Nepal, Bhutan, and northwestern India (Figure 1; *Vance et al.*, 2003; *Wobus et al.*, 2005; *Finnegan et al.*, 2008; *Godard et al.*, 2012, 2014; *Lupker et al.*, 2012, 2017; *Munack et al.*, 2014; *Puchol et al.*, 2014; *Scherler et al.*, 2014b; *West et al.*, 2015; *Morell et al.*, 2015a, 2017, *Olen et al.*, 2015, 2016; *Portenga et al.*, 2015; *Dietsch et al.*, 2015; *Le Roux-Mallouf et al.*, 2015; *Abrahami et al.*, 2016; *Adams et al.*, 2016; *Kim et al.*, 2017;
*Dingle et al.*, 2018). These rates are broadly consistent with geographic patterns found elsewhere along the range, with the fastest denudation in the core of the range.

        In Far Western Nepal, there are few measurements of erosion rates on other timescales that our millennial-scale denudation rates can be compared to. As far as we are aware, the lone fluvial sediment flux measurement that is close to our sites is on the Karnali River and was collected within 10 km of our Karnali Ferry sampling site (*Andermann et al.*, 2012). The
suspended sediment flux of 492 t km$^{-2}$ yr$^{-1}$ measured at this site is based on a rating curve of suspended sediment concentration against water discharge derived from five years of measurements (1973-74, 1977-79) and applied to 34 years of water discharge measurements (1973-2006). Because this does not include bedload or solute fluxes (which were not measured at this site), it underestimates the total fluvial mass flux by an unknown amount. Andermann et al. (2012) noted that bedload fluxes in other mountainous rivers typically range from 2 to 40% of the total mass flux (*Turowski et al.*, 2010) and that solute fluxes in
comparable Himalayan rivers typically range from 1 to 4% of the total flux (e.g., *Summerfield and Hulton*, 1994; *Gabet et al.*, 2008), such that the total fluvial mass flux may feasibly be ~50% larger than the suspended sediment flux, or ~740 t km$^{-2}$ yr$^{-1}$. This range of 492 – 740 t km$^{-2}$ yr$^{-1}$ is roughly a factor of 1.5-2 lower than our $^{10}$Be-derived denudation rate at this site (1717 ± 157 t km$^{-2}$ yr$^{-1}$), which, as a rate inferred from cosmogenic nuclides, includes all mass fluxes from the upper few meters below the Earth's surface (including material that ultimately becomes bedload and solutes in rivers), which is where cosmogenic
nuclides are predominantly produced (e.g., *Riebe et al.*, 2003; *Dixon et al.*, 2009a; *Ferrier et al.*, 2010). This difference of a

factor of 1.5-2 is relatively small compared to the order-of-magnitude differences between short-term and long-term rates often observed in small catchments (e.g., *Kirchner et al.*, 2001; *Hewawasam et al.*, 2003; *Covault et al.*, 2013). To the extent that this basin is representative of our other sampled basins, this is consistent with relatively small differences between decadal-scale and millennial-scale mass fluxes in the study region.

On million-year timescales, exhumation rates can be inferred from thermochronometric measurements (e.g., *Thiede and Ehlers*, 2013). Relatively few thermochronometric measurements, however, have been made in Far Western Nepal relative to in central and eastern Nepal, Bhutan, and northern India. Robinson et al. (2006) measured $^{40}Ar/^{39}Ar$ muscovite cooling ages from the Ramgarh thrust zone in the Greater Himalayan and Tethyan sequences to show that this major fault zone was active at ~12 Ma, but did not report exhumation rates. Recently, van der Beek et al. (2016) used apatite fission track measurements

to infer Myr-scale exhumation rates of ~0.5 to 2.5 mm yr$^{-1}$ at a series of points along the Karnali River, equivalent to ~1325 to 6625 t km$^{-2}$ yr$^{-1}$ for a bedrock density of 2650 kg m$^{-3}$. These exhumation rates are consistent with the inferred denudation rates of $1717 \pm 157$ to $2044 \pm 209$ t km$^{-2}$ yr$^{-1}$ in our two Karnali River samples (Table 2), and hence consistent with a similarity in erosion rates over millennial to million-year timescales.

**7. Implications for controls on denudation rate**

      The relative importance of various controls on denudation in the Himalaya is a longstanding matter of interest and contention. In central Nepal, Burbank et al. (2003) observed no systematic variations in thermochronometrically-inferred exhumation rates across a five-fold range of monsoon precipitation rates, and interpreted this as an indication that tectonically driven rock uplift is the primary driver of Himalayan erosion. Similarly, recent studies used cosmogenic-based denudation

rates in central Nepal (Godard et al., 2014), eastern Nepal (Olen et al., 2015), and the Garhwal Himalaya in northern India (*Scherler et al.*, 2014) to infer that Himalayan erosion is mainly limited by rock uplift rate and is only secondarily sensitive to climatic forcings. In contrast, others have proposed that climate exerts a strong control on Himalayan denudation. Thiede et al. (2004), for example, proposed that climatically controlled surface processes are the primary control on exhumation rates inferred from apatite fission track measurements in the Himalayan crystalline core of the Sutlej basin. Recently, Olen et al.

(2016) used a compilation of cosmogenic-based denudation rates across the Himalaya to argue that while tectonic drivers are the dominant control on the mean denudation rate, climate and vegetation control the regional variance in denudation rates.

      On their own, our measurements cannot resolve the relative strengths of tectonic and climatic controls on denudation in Far Western Nepal, though they are consistent with a sensitivity of denudation rate to both tectonics and climate. For example, our inferred denudation rates and $k_{sn}$ values are largely consistent with a compilation of previous measurements

across the Himalaya (Figure 5A; Scherler et al., 2017). This implies a similar relationship between denudation rate and channel longitudinal profiles in Far Western Nepal as at the other sites in the Himalaya where these have been measured. Following the interpretation of Scherler et al. (2014), we note that the positive nonlinear relationship between denudation rate and $k_{sn}$ across the Himalaya-wide dataset is consistent with a dominantly tectonic control on denudation rate. This interpretation is supported by the patterns of microseismicity in the study basins, which are broadly consistent with stronger tectonic forcings

in the basins with higher inferred denudation rates (Tables 1-2).

      Our inferred denudation rates are positively correlated with basin-average specific stream power, though with substantial scatter ($R^2 = 0.56$; Figure 5B). These are consistent with measurements of denudation rate and stream power elsewhere in the Himalaya (*Olen et al.*, 2016; Figure 5B), and are therefore consistent with Olen et al.'s (2016) interpretation that Himalayan denudation is dominantly tectonically driven and secondarily modulated by climate. Thus, the positive

relationships between denudation rate and both $k_{sn}$ and specific stream power data suggest that denudation rate in Far Western Nepal is sensitive to both tectonic and climatic forcings, though the dataset is insufficiently large to isolate the relative importance of each forcing in this region.

      Two of our measurements suggest a path toward a better understanding of the relative importance of tectonic and climatic drivers of erosion in Far Western Nepal. One of our study basins, Raduwa, has an inferred denudation rate several

times lower than those in the other study basins ($385 \pm 31$ t km$^{-2}$ yr$^{-1}$; Table 2). Climatic factors do not appear to be the dominant cause of slow erosion in Raduwa. Mean annual precipitation in Raduwa exceeds 1800 mm, third highest amongst our study basins, such that if erosion were dominantly climatically controlled, one would expect Raduwa's denudation rate to be several times higher, comparable to that in the basins with similar precipitation rates (Tables 1-2). Likewise, basin-averaged specific stream power values in Raduwa are relatively low but not the lowest among our study basins (Table 1), implying that

stream power alone cannot explain the low denudation rate. By contrast, several observations indicate that Raduwa has the

lowest tectonic forcing among our study basins. Raduwa has the lowest mean $k_{sn}$, lowest mean hillslope gradient, and one of the lowest rates of seismicity per unit area among the study basins (Table 1). The entire Raduwa basin lies south of the southern physiographic transition (PT2-S) identified by *Harvey et al.* (2015), and thus south of the zone where crustal exhumation is expected to be fastest. In this respect it is unlike the other study basins, all of which either lie northeast of PT2-S, where exhumation is expected to be faster, or span both sides of it. This is consistent with findings in central Nepal and the Garhwal Himalaya, where erosion is slower and topography is gentler south of the lone physiographic transition, PT2 (*Wobus et al.*, 2005; *Scherler et al.*, 2014a). We interpret the low denudation rate in Raduwa to be consistent with slower rock uplift in Raduwa than in the other study basins, and thus a reflection of a tectonic influence on denudation rate. Further measurements in the lower-elevation terrain southwest of this physiographic transition will be needed to fully test this hypothesis.

At the other end of the spectrum is the Budhiganga basin, whose inferred denudation rate ($8737 \pm 2908$ t km$^{-2}$ yr$^{-1}$) is several times higher than those in the other study basins (Table 2). It is also several times higher than denudation rates in basins with similar basin-average $k_{sn}$ elsewhere in the Himalaya (Table 1), implying that its denudation rate is unusually high for the steepness of its channels. It is not clear what is responsible for the high inferred denudation rate at Budhiganga. Among our study basins, Budhiganga has among the highest $k_{sn}$, hillslope gradient, seismicity per unit area, mean annual precipitation, and stream power, suggesting that both tectonic and climatic forcings are relatively strong in the Budhiganga basin (Table 1). None of these metrics, however, are exceptionally high relative to the other study basins. Indeed, the Kalanga basin, which is west of Budhiganga at a similar elevation (Figure 3), has similar values for these same metrics (Table 1) but an inferred denudation rate $4.8 \pm 1.7$ times lower than that for Budhiganga. We are unaware of measurements of the erodibility of each lithology in these basins, but current geologic maps suggest the difference between the inferred Budhiganga and Kalanga denudation rates is unlikely to reflect a lithologic control. The Budhiganga basin is dominated by crystalline rocks (migmatitic and calc-silicate gneiss, metavolcanics, orthogneiss, schist, and granite; Figure S1; *Gansser*, 1964; *Arita et al.*, 1984; *Upreti and Le Fort*, 1999; *DeCelles et al.*, 2001; *Robinson et al.*, 2006), which tend to be relatively strong and inhibit rapid erosion, while the Kalanga basin is dominated by carbonates, which tend to be less resistant to erosion (*Sklar and Dietrich*, 2001). Thus, if denudation rates were dominantly controlled by lithology, then to the extent that the mapped lithologic units have erodibilities similar to those of analogous rocks in laboratory tests (e.g., *Sklar and Dietrich*, 2001), denudation in the Kalanga basin should be faster than that in the Budhiganga, not slower. Evaluating lithologic effects on denudation rate in the study area more rigorously will require new detailed geologic mapping and new measurements of lithologic characteristics (e.g., tensile strength, fracture spacing, bedding thickness) in these units. We cannot rule out the possibility that the inferred denudation rate is related to landslide-derived sediment in the Budhiganga sample, although we do not have independent evidence that indicates that this sample contains high concentrations of landslide-derived sediment (Section 5.1; Figure S6). Indeed, the sample collected at Budhiganga contains <1% by mass of grains larger than 2 mm in size, which is inconsistent with a dominantly landslide-derived source of sediment and comparable to that in most of our other samples (Table S1).

While our measurements are unable to isolate the controls on denudation rates on their own, they are nonetheless able to highlight regions in Far Western Nepal that may be useful targets for future denudation rate measurements. For example, the regions northeast of 100 km and southwest of 180 [KF1]km in the swath in Figure 2 have few denudation rate measurements, and targeting small basins within these regions may be particularly instructive. For example, to isolate the sensitivity of denudation rates to rock uplift rate, it may be useful to target basins like Raduwa in the Subhimalaya and the southwestern portion of the Lesser Himalaya (Figure 3A), where topographic relief and rock uplift rates are hypothesized to be low, to determine whether the low denudation rate in Raduwa is typical of other basins undergoing similar tectonic forcing. Similarly, it may be helpful to target small catchments that lie over the hypothesized mid-crustal ramps (*Harvey et al.*, 2015), where rock uplift rates may be higher, including small basins at relatively high elevation in the Greater Himalaya within the Karnali, Bheri, and Seti River basins (Figure 3B). Likewise, to assess the sensitivity of denudation rates to stream power, it may be useful to target small basins that isolate regions of high SSP, like those at mid-elevations in the Bheri catchment and high elevations in the Seti catchment, as well as basins that isolate regions of low SSP, like those at low elevation in the southwestern portions of the Bheri, Karnali, and Seti basins (Figure 4C). While some of these sites are relatively remote and were beyond our ability to access them during this study, future field campaigns that are able to collect samples from these sites may be able to put stricter constraints on the couplings between denudation rate, rock uplift, and climate in Far Western Nepal.

**8. Conclusions**

The primary contribution of this study is a new suite of millennial-scale basin-average denudation rates inferred from cosmogenic [10]Be concentrations in stream sediment in the Himalayas of remote Far Western Nepal. The inferred denudation rates, normalized channel steepness, and specific stream power in the study basins are largely consistent with previous measurements elsewhere in the Himalaya. These measurements represent a first step toward filling an important gap in denudation rate measurements in Far Western Nepal, which may be useful for testing the sensitivity of denudation rate to along-strike variations in several quantities (e.g., fault geometry, stream power), and illustrate the need for future denudation rate measurements in the region to test hypotheses about feedbacks between climate, tectonics, and topography across the Himalaya.

## Acknowledgments

We thank Cliff Riebe and Darryl Granger for laboratory advice and assistance during sample preparation, Hella Wittmann-Oelze for providing low-background beryllium carrier solution, Kim Cobb and Hussein Sayani for assistance measuring Al concentrations in quartz, and Bodo Bookhagen and Greg Ruetenik for helpful discussions regarding stream power in the Himalaya.

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

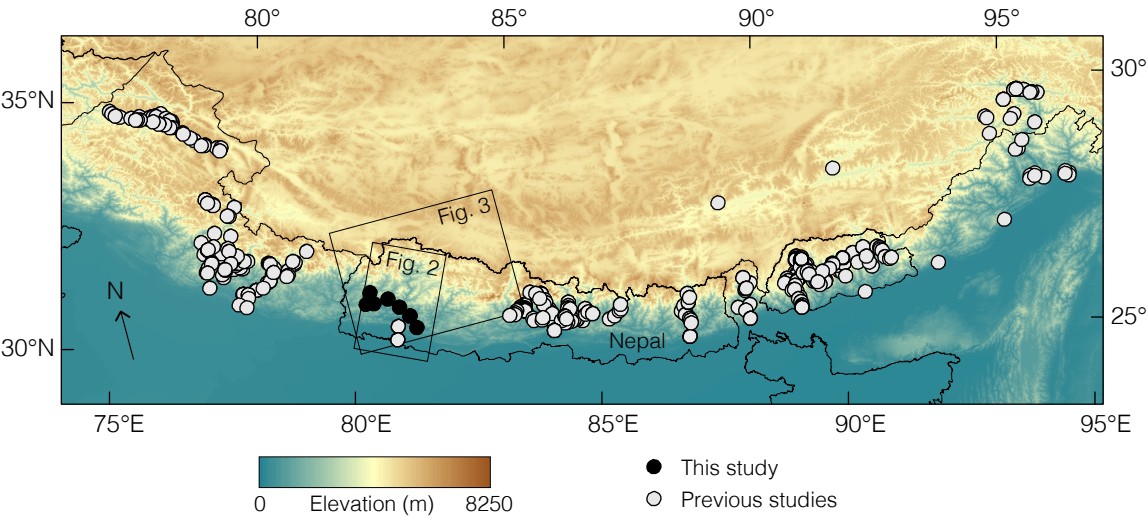

**Figure 1.** The relative paucity of denudation rate measurements in western Nepal is shown by the sampling locations of basin-average denudation rates inferred from cosmogenic nuclide measurements in previous studies (gray circles; *Vance et al.*, 2003; *Wobus et al.*, 2005; *Finnegan et al.*, 2008; *Godard et al.*, 2012, 2014; *Lupker et al.*, 2012, 2017; *Munack et al.*, 2014; *Puchol et al.*, 2014; *Scherler et al.*, 2014; *West et al.*, 2015; *Morell et al.*, 2015, 2017; *Olen et al.*, 2015, 2016; *Portenga et al.*, 2015; *Dietsch et al.*, 2015; *Le Roux-Mallouf et al.*, 2015; *Abrahami et al.*, 2016; *Adams et al.*, 2016; *Kim et al.*, 2017; *Dingle et al.*, 2018. Sampling locations of new denudation rate measurements in Far Western Nepal in this study are shown as black circles. Background colors show elevation in SRTM topographic data (*Farr et al.*, 2007). Rectangles show the location of the swath profile in Figure 2 and the study region in Far Western Nepal in Figure 3.

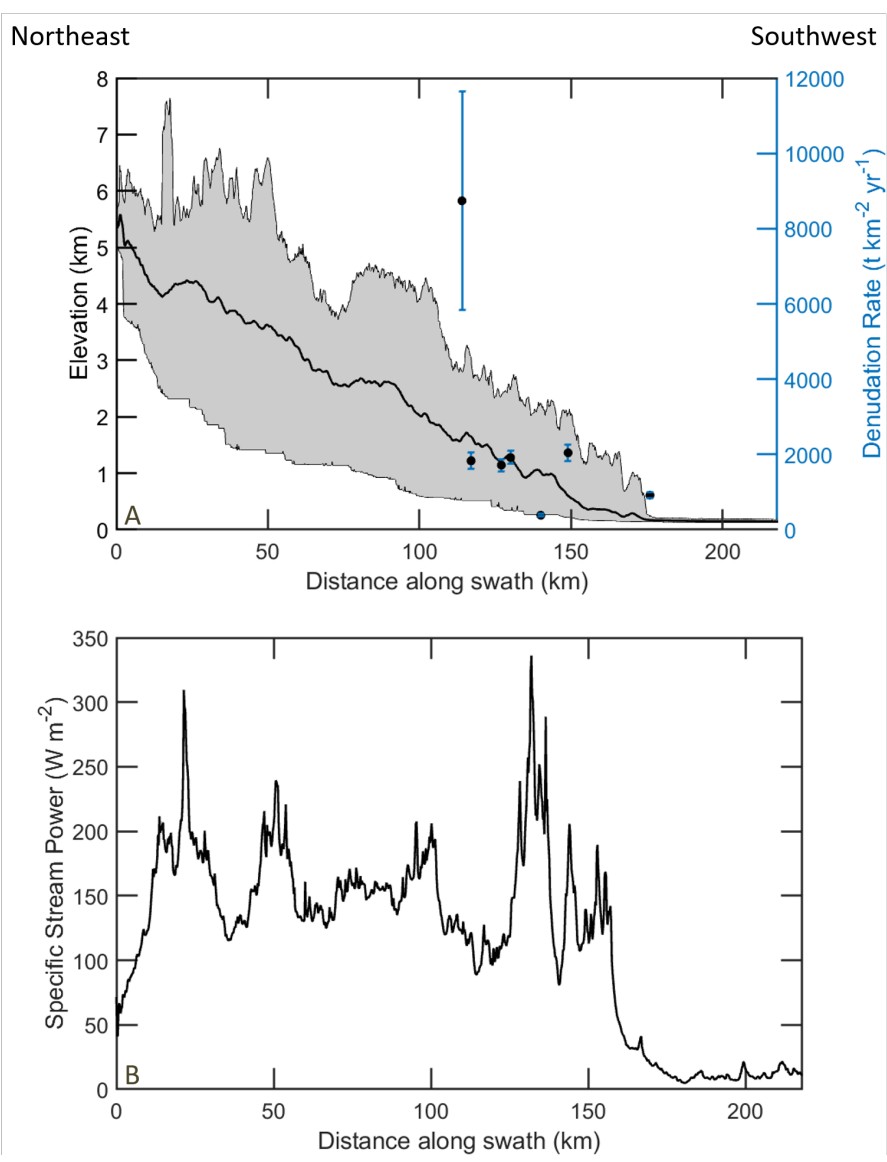

5  **Figure 2A.** Profiles of mean elevation (black line) and range of minimum to maximum elevations (gray shaded region) perpendicular to the Himalayan range front along the swath shown in Figure 1. Black circles show denudation rate estimates (Table 2) and locations of sediment samples. **B.** Profile of mean specific stream power along the same region.

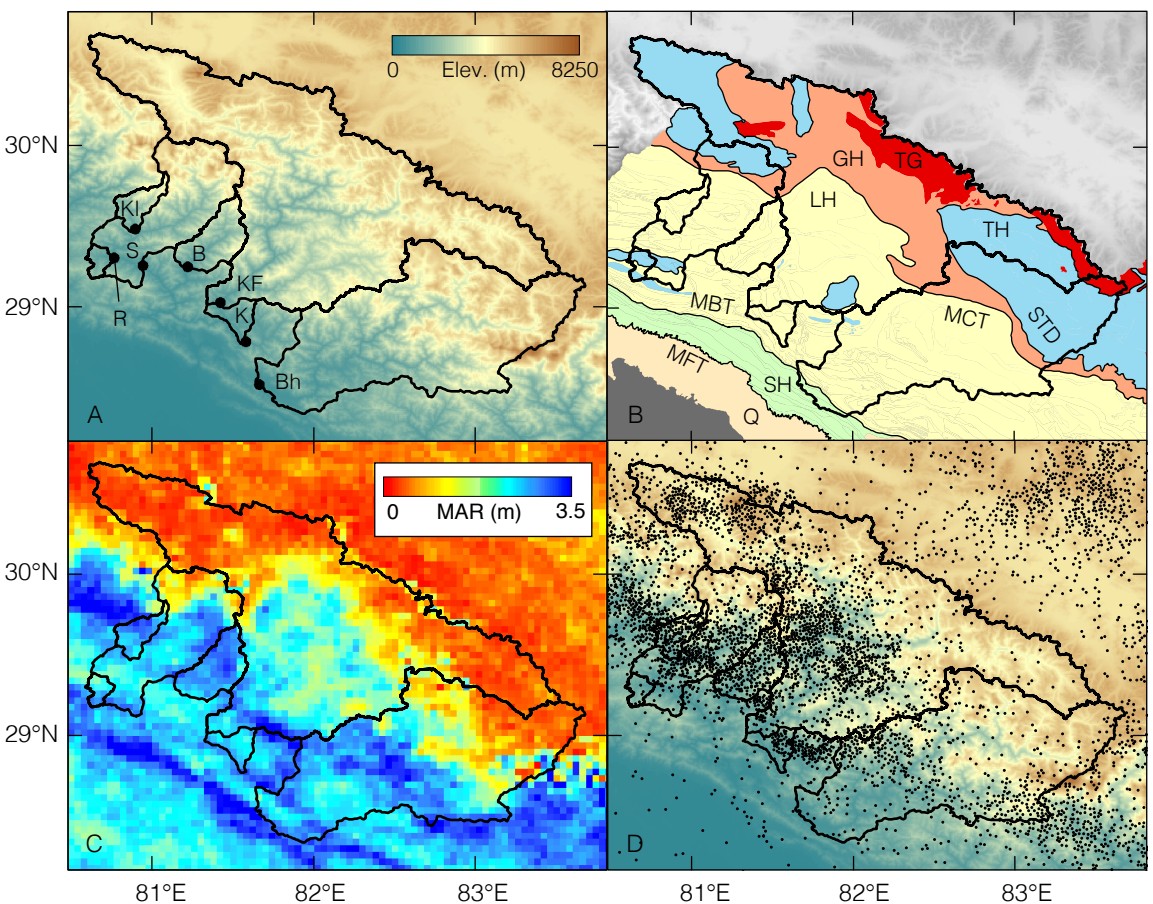

**Figure 3A.** Stream sediment sampling locations (circles) and upstream drainage basins (black lines) over which the inferred denudation rates are averaged (Section 3.1). Here letters refer to the sample basin names: B = Budhiganga, Bh = Bheri, K = Karnali, KF = Karnali Ferry, Kl = Kalanga, R = Raduwa, and S = Seti. **B.** Geologic map of the major tectonostratigraphic units in the Himalaya of Far Western Nepal (compiled from *DeCelles et al.*, 2001; *Robinson et al.*, 2006; *Yin*, 2006; *Ojha et al.*, 2009). Here GH = Greater Himalaya, LH = Lesser Himalaya, SH = Subhimalaya, TG = Tertiary granites, TH = Tibetan Himalaya, Q = Quaternary, MBT = Main Boundary Thrust, MCT = Main Central Thrust, MFT = Main Frontal Thrust, and STD = South Tibetan Detachment. **C.** Mean annual rainfall in meters (MAR; Bookhagen and Burbank, 2010). **D.** Locations of $M_w > 2$ microseismic events (compiled from the epicenter map published by the Department of Mines and Geology of Nepal, 1992-2005).

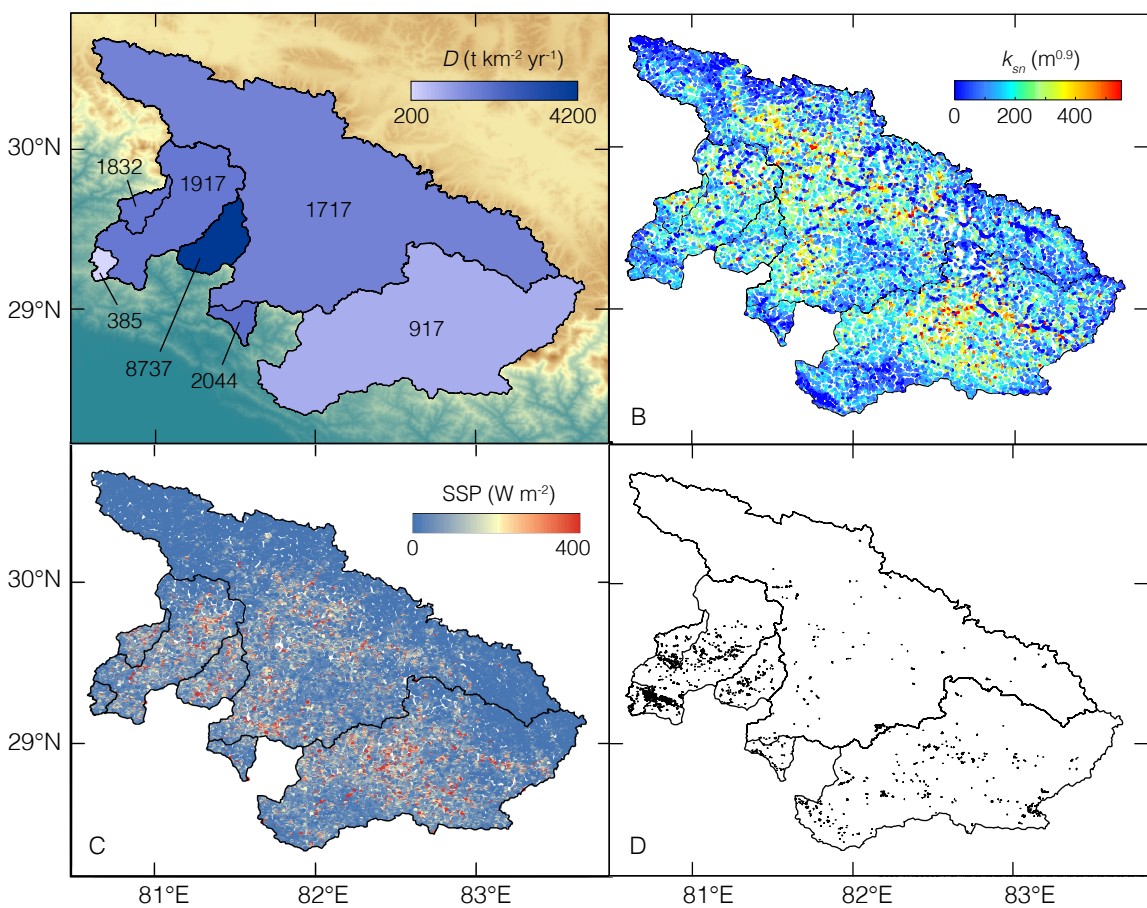

**Figure 4A.** Mean denudation rates inferred from measured [10]Be concentrations. Background color shows elevation as in Figure 3A. **B.** Normalized channel steepness, $k_{sn}$ (Equation 1). **C.** Specific stream power (Section 3.4). **D.** Locations of mapped landslide scars (Section 3.5).

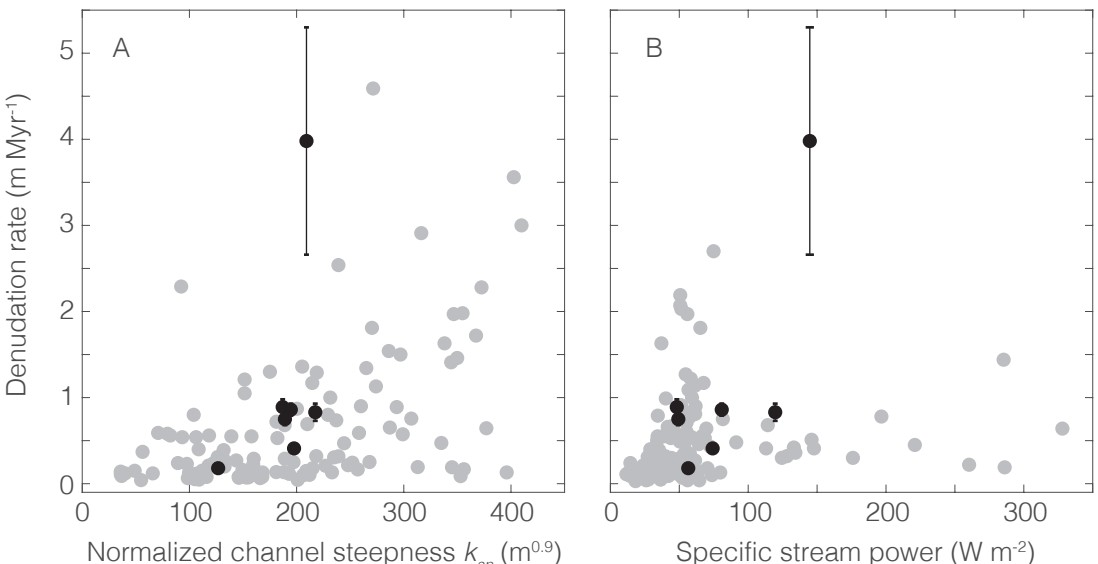

**Figure 5.** Basin-averaged denudation rates in the Himalaya versus normalized channel steepness (panel A) and specific stream power (panel B). In both panels, black dots indicate our study sites in Far Western Nepal and gray dots indicate sites elsewhere in the Himalaya (Scherler et al. (2017) and Adams et al. (2016) in panel A, and Olen et al. (2016) in panel B). To avoid introducing biases to comparisons of denudation rate estimates determined from different versions of the CRONUS calculator, the black dots in Figure 5 show denudation rate estimates at our study sites that have been recalculated using the same version of the CRONUS calculator (v2.2) as that used in Adams et al. (2016), Olen et al. (2016), and Scherler et al. (2017). These rates are an average of 19% (range: 16-26%) higher than those calculated with CRONUS v2.3 in Table 2.

**Table 1.** Stream sediment sampling sites and upstream basin characteristics.

Sampling locations      Basin characteristics

| Name | Lat. (°N) | Lon. (°E) | Area (km$^2$) | Bedrock units[a] | $f_{ice}$[b] (%) | $f_{scar}$[c] (%) | Mean lat. (°N) | Mean lon. (°E) |
|---|---|---|---|---|---|---|---|---|
| Bheri | 28.51761 | 81.66444 | 14,507 | SH, LH, GH, TH | 5.1 | 0.08 | 28.840 | 82.640 |
| Budhiganga | 29.24682 | 81.21924 | 1399 | LH | 0.5 | 0.37 | 29.414 | 81.390 |
| Kalanga | 29.47964 | 80.89717 | 634 | LH | 1.7 | 0.31 | 29.716 | 80.917 |
| Karnali | 28.78066 | 81.57983 | 24,565 | LH, GH, TH | 5.7 | 0.03 | 29.734 | 81.988 |
| Karnali Ferry | 29.02343 | 81.42153 | 24,052 | LH, GH, TH | 5.9 | 0.03 | 29.633 | 81.999 |
| Raduwa | 29.30191 | 80.76624 | 237 | LH, TH | 0.0 | 0.42 | 29.285 | 80.671 |
| Seti | 29.25429 | 80.74370 | 5438 | LH, GH, TH | 6.9 | 0.20 | 29.587 | 81.067 |

Basin-averaged values

| Name | $k_{sn}$[d] (m$^{0.9}$) | Mean annual rainfall[e] (mm) | Specific stream power[f] (W m$^{-2}$) | Hillslope gradient (°) | Microseismicity per unit area[g] (events km$^{-2}$) |
|---|---|---|---|---|---|
| Bheri | 198 | 1478 | 74.2 | 22.6 | 0.033 |
| Budhiganga | 209 | 2289 | 144.7 | 23.1 | 0.149 |
| Kalanga | 217 | 2287 | 119.6 | 25.3 | 0.189 |
| Karnali | 187 | 977 | 48.2 | 21.2 | 0.064 |
| Karnali Ferry | 189 | 953 | 49.3 | 21.2 | 0.063 |
| Raduwa | 127 | 1850 | 56.5 | 16.7 | 0.038 |
| Seti | 195 | 1640 | 80.8 | 23.4 | 0.109 |

[a] Abbreviations for tectonostratigraphic units: GH = Greater Himalaya, LH = Lesser Himalaya, SH = Subhimalaya, TH = Tibetan Himalaya (Section 2).
[b] Fraction of basin covered by glacial ice (Section 3.2.1).
[c] Fraction of basin mapped as landslide scars (Section 3.5).
[d] Normalized channel steepness (Section 3.3).
[e] From Bookhagen and Burbank (2010) (Section 3.4).
[f] Specific stream power computed as in Section 3.4.
[g] Number of $M_w > 2$ events per unit area during 1992-2005 (Figure 3D).

**Table 2.** Cosmogenic $^{10}$Be concentrations and inferred denudation rates computed first under only topographic shielding and second under the combination of topographic and seasonal snow shielding.

| Name | Quartz mass (g) | $^{9}$Be spike $(10^{-6}$ g) | $^{10}$Be/$^{9}$Be $(10^{-15})$ (mean ± unc.) | $N$ ($^{10}$Be conc.) (atoms g$^{-1}$) | $N_{ice\text{-}free} = N(1-f_{ice})^{-1}$ (atoms g$^{-1}$) |
|---|---|---|---|---|---|
| Bheri | 117.682 | 257.1 | 316.1 ± 11.28 | 40,705 ± 1491 | 42,892 ± 1571 |
| Budhiganga | 153.449 | 251.1 | 43.03 ± 10.76 | 3612 ± 1067 | 3630 ± 1072 |
| Kalanga | 59.138 | 256.8 | 79.67 ± 6.676 | 19,155 ± 1775 | 19,486 ± 1806 |
| Karnali | 136.057 | 255.5 | 246.5 ± 15.01 | 27,127 ± 1702 | 28,766 ± 1805 |
| Karnali Ferry | 154.388 | 247.0 | 346.7 ± 14.58 | 32,758 ± 1409 | 34,812 ± 1497 |
| Raduwa | 133.077 | 262.1 | 418.1 ± 12.09 | 48,790 ± 1441 | 48,790 ± 1441 |
| Seti | 134.001 | 256.5 | 166.9 ± 6.799 | 18,487 ± 796 | 19,857 ± 855 |
| Process blank | n/a | 268.4 | 6.338 ± 1.269 | n/a | n/a |

| Name | Shielding factors[a] | | | Only topographic shielding | | |
|---|---|---|---|---|---|---|
| | Topographic | Snow | Total | Basin-avg. $P$[b] (atoms g$^{-1}$ yr$^{-1}$) | Effective elevation (m) | Denudation rate (t km$^{-2}$ yr$^{-1}$) |
| Bheri | 0.9788 | 0.8727 | 0.8536 | 34.4266 | 3575 | 1338 ± 119 |
| Budhiganga | 0.9809 | 0.9758 | 0.9572 | 19.3480 | 2580 | 9382 ± 3124 |
| Kalanga | 0.9759 | 0.9535 | 0.9300 | 22.9986 | 2826 | 2007 ± 246 |
| Karnali | 0.9818 | 0.7971 | 0.7817 | 48.5659 | 4253 | 2837 ± 293 |
| Karnali Ferry | 0.9816 | 0.7925 | 0.7770 | 49.4950 | 4250 | 2340 ± 217 |
| Raduwa | 0.9939 | 0.9984 | 0.9924 | 10.4955 | 1559 | 385 ± 31 |
| Seti | 0.9793 | 0.9190 | 0.9001 | 27.0919 | 3106 | 2300 ± 209 |

| Name | Topographic and snow shielding | | |
|---|---|---|---|
| | Basin-avg. $P$[b] (atoms g$^{-1}$ yr$^{-1}$) | Effective elevation (m) | Denudation rate (t km$^{-2}$ yr$^{-1}$) |
| Bheri | 26.5205 | 3101 | 917 ± 80 |
| Budhiganga | 18.3604 | 2494 | 8737 ± 2908 |
| Kalanga | 20.8686 | 2742 | 1832 ± 224 |
| Karnali | 35.2527 | 4024 | 2044 ± 209 |
| Karnali Ferry | 35.8678 | 4070 | 1717 ± 157 |
| Raduwa | 10.4791 | 1559 | 385 ± 31 |
| Seti | 22.3340 | 2916 | 1917 ± 173 |

[a] Mean shielding factors for ice-free regions in each basin.
[b] Basin-averaged $^{10}$Be production rate.

