# Peer review of "Millennial-scale denudation rates in the Himalaya of Far Western Nepal"

_Earth Surface Dynamics, 2019_

## Referee Comment (RC1) · Anonymous Referee #1 · 28 Mar 2019

The paper by Lujendra Ojha, Ken L. Ferrier and Tank Ojha presents seven 10Be-derived estimates of catchment-scale denudation rates in Western Nepal. These denudation rates fall within the range of values published elsewhere in Himalaya. The paper is clear, well written and illustrated.

It seems that the main interest of this contribution is to provide new denudation data in a region where they were lacking. The authors do not claim to revolution the debate between the climatic and tectonic control of denudation with their data, which is fair, but the scientific justification of the study is a bit short. Why was it "important" to fill this gap of data? Is it a key place? Was the sampling initially designed for some particular question? The last sentence of the conclusion says that this study "illustrates the need for future denudation rate measurements in the region to test hypotheses". As it, I

am not sure that this is the case. I agree that new data are always useful and that is why I think that this paper should be published. Nevertheless, this study shows that these new denudation data are not able to discuss the relative contribution between climate and tectonics. Thus where do we need to gather future samples to answer this question and why?

The calculation of the denudation rates is correct and includes a good discussion of uncertainties. Yet, some aspects of these uncertainties could be improved:

Dibiase (Earth Surf. Dynam., 6, 923-931, 2018) recently showed that the topographic shielding correction is usually unappropriated. As denudation values with and without topographic shielding are already given, the authors could only recall the Dibiase's paper. The uncertainty on production rate is not indicated. It can easily reach more than 10% and depends on the production rate model (which one was used in the CRONUS calculator?) and thus should be propagated to the denudation rate uncertainty.

What is exactly the maximum grain size of quartz that was dissolved? Table S1 gives the distribution of grain sizes for each sample. I understand that all these grain sizes were dissolved together. Why doing this, while many studies have shown that the 10Be concentration is grain size dependent? The grain size distribution differs a lot between catchments. Does denudation rate correlate with the mean (or other metrics) grain size in this dataset, as observed in many other cases?

The lithological effect is nicely discussed by exploring two end-members scenarios where only the catchment head or the catchment foot provides quartz. However, how do the relationships between denudation and slope, steepness and stream power change when restricted to the upper or lower catchment parts? For example, by restricting the calculation of ks and mean stream power to the upper Budhiganga catchment, its "anomalous" high denudation may be shifted to the right on diagrams of Figure 4, in a more "classical" configuration. In the studied catchments there is a correlation between lithology (possibly quartz content) and elevation (10Be production rate), as

elsewhere along the range. Could it be possible that the lithological effect explains the large dispersion observed between denudation and steepness (see for example Carretier et al., Earth Surface Processes and Landform, 2015)?

The discussion on the erodibility of different rocks in the Budhiganga and Kalanga catchments, ruling out lithology as possible control of their different denudations, is maybe a bit short (paragraph 20). Are the crystalline rocks of the Budhiganga catchment weathered or fractured? Are the carbonates of the Kalanga layered and possibly more easy to erode? What do other studies in the region or close say about the lithological control of denudation (e.g. Lave and Avouac, JGR, 2001)? Furthermore, this discussion may change by restricting the calculation of ks and stream power to the quartz-rich lithology (previous comment).

I feel that the answers to these comments are quite straightforward. I encourage the authors to add the suggested analysis.

---

## Referee Comment (RC2) · Anonymous Referee #2 · 29 Mar 2019

This article by Ojha and co-authors presents a new dataset of basin-wide denudation rates derived from 10Be measurements in rivers sands. The dataset consists in 7 new 10Be concentration measurements for the Far Western Nepal region in the Himalayas. While numerous studies have reported CRN derived denudation rates in various parts of the Himalayan arc, this is one of the first dataset from this particular area. The data and the methods are well presented and the authors discuss thoroughly the various hypothesis and caveats when calculating the denudation rates from the 10Be concentration measurements. They compare their results with available data in other areas along the Himalayan arc, and then discuss the relative contributions of various types of forcings to denudation rates. Seven samples is quite a small dataset when compared with other similar studies in this area. However, this is a very important and understud-

[Figure]

**ESurfD**

Interactive
comment

ied part of the arc (as noted by the authors this area is difficult to access). Indeed, the region illustrates the existence of important along-strike variations, in particular with respect to the intensively documented central Nepal sections. Any new data is thus a very welcome addition to the body of knowledge of the dynamics of the Himalayas, and has the potential to provide critical constraints on future discussion of the lateral variability along the arc.

My main concern is that, in its present form, the article lacks the formulation of a clear problem statement. The abstract and in particular the introduction read like the primary focus of the article is just to present a new dataset (see for example line 8), which is not a very attractive prospect for potential readers. For example, recent studies highlight a number of peculiar features of Far West Nepal (Harvey et al., 2015, van der Beek et al., 2016) and it might provide a starting point to present the results in terms of the analysis of lateral variations along the arc, in particular with respect to the much better constrained central Nepal area. Concerning the discussion of the implications on the effective controls on denudation, I feel that figure 4 only provides a very blurred and generalized perspective on the problem, and is not a very robust support for this discussion at the scale considered here. Comparing cross sections for denudation data, topography, geology and precipitation (etc . . .) in Far West and Central Nepal, could be very helpful for that purpose. See for example figure 2 of van der Beek et al. (2016).

Specific comments keyed to line numbers

p1-l38 : at this stage of the introduction you should highlight clearly why this region is important and interesting.

p2-l1-8 : the few studies that have looked in details at this area (van der Beek et al., 2016 and Harvey et al., 2015) have articulated a clear problem statement, and you could build on that to reformulate your introduction.

p2- l30-33 : the presentation of the ksn belongs to the Methods section.

[Figure]

section 2 Provide some information about the implication of these lithological variations between the different units in terms of relative quartz abundance.

p3 - l24 what is the upper limit for the grain size analyzed?

p3- l35 : this is discussed later, but you should clearly state the fact that you do not take into account the variations in quartz content as a potential limitation.

P5 l20-23 : sentence not clear, "why most applicable?"

P5 l40-45 : this is interesting and not frequent in this kind of studies, so it might be worth giving a bit more visibility to the corresponding results.

section 5.1 : in the absence of indications on the grain size used here it is difficult to discuss the impact of eventual landsliding contribution on the measured concentration.

P7 l5-20 I would be surprised if the contribution of chemical weathering were significant in this tectonic context and at this scale. I think there are estimates of solutes fluxes in one of the Lupker et al. paper.

P8 l25 "larger tan 250 microns" is not a very precise definition of the grain size and does not allow to make a robust claim on this point.

section 6 - even if you have less data it might be interesting, in terms of comparison, to plot you denudation rates along a cross section perpendicular to the range (as well as topography), with similar sections for central and/or eastern Nepal.

section 7 - same as previous points, having cross sections with denudation, topography, precipitation (+ thermochron data, etc. . .), might help to put everything into context, and make the argument easier.

Figure 3B&C : at this scale there might be some overlaps with the data points representing ksn or Spw on individual stream segments, it would probably be better to use a continuous representation (topotoolbox has some function for that purpose). Same comment for S2 and S4.

Figure 4 : The trends and relationships discussed in the text should be plotted on the figure with their confidence intervals. Figure 2b of Scherler et al (2014) display less scatter than what you plot here, did you subset the data according to some criteria (glaciated, lithology, ....)? You could display the trends identified by Olen et al. in adjacent regions. A and B are actually not indicated on the figure.

---

## Referee Comment (RC3) · Maarten Lupker (Referee) · 2 Apr 2019

This manuscript adds new cosmogenic 10Be data to the growing number of studies mapping out modern denudation rates across the Himalayan arc and fills a data-gap in Western-Nepal where no such data was published so-far. Calculated denudation rates are at par with other central-Himalayan catchments and the authors discuss it in the context of tectonic and climatic driving processes.

This data-set is relatively modest in size (7 samples) and the study is not very ambitious in the way it is set up. However, it provides new and useful data that is worth publishing in my opinion. The authors provide a very detailed and careful discussion of the limitation of the cosmogenic nuclide-derived denudation approach, carefully evalu-

ating the different steps which is something I have appreciated. The writing is clear and the figures are informative. I, however, think that the rationale for the study could be improved so that it reads less like a simple data-report (which is not a bad thing per se, but in that case, an e-surf research article might not be the most appropriate format).

As it currently stands, the authors are focusing on the climate vs. tectonics debate to motivate their study. But I feel that with a limited number of additional samples and a relatively short analysis they do not contribute much to this discussion. This would require a more thorough re-analysis of all 10Be along the Himalayan arc than what is presented here. On the other hand, the authors could have focused in more details on the specificities of the Karnali catchment with older AFT ages and lower stream power values compared to other parts of the range (e.g. van der Beek et al., 2016 – Geology). How these characteristics may or may not be expressed in surface catchment denudation rates seems a worthy discussion angle for this manuscript that is not necessarily very well addressed in this submission.

Some more detailed considerations:

- It seems that the authors try to provide a global overview of 10Be data available across the Himalayan range (Figure 1). This is useful but is incomplete. At least 4 papers (maybe more) were omitted and should be mentioned: Puchol et al., 2014 – Geomorphology; West et al., 2015 – Esurf; Lupker et al., 2017 – Esurf; Dingle et al., 2018 – Esurf.

- The comparison between this dataset and published data (e.g. Figure 4) should be made carefully. If I understand it correctly the denudation rate, as well as steepness or stream power, have not been recalculated in a homogeneous way for different datasets. Given that the authors report some large variations between different approaches (e.g. snow cover effect) it is uncertain how these differences may bias this type of comparison. I would suggest to recalculate the data using a homogeneous procedure (even though I am aware that this represents a significant amount of work) or convincingly

show that the differences are minor.

- The use of a topographic shielding correction in catchment-wide denudation rates has been recently questioned DiBiase, 2018 – Esurf

- p.6, l.52: it is not clear to me how grain-size data on fluvial sediments will tell you much about the importance of landslide inputs given transport segregation processes.

- On the effect of chemical erosion on 10Be denudation estimate (p.7, section 5.2): the fact that chemical denudation is only a very small fraction of the overall mass export (as mentioned later in the manuscript) should provide a rough estimate on the magnitude of this bias. - p.8, l.26: Puchol et al., 2015 – Geomorphology provides a direct example of 10Be concentrations correlated with grain-size induced by landslide processes in a Himalayan catchment.

- p.9, l. 1-2 the difference between short-term denudation estimates and long-term rates in the Himalaya has been very recently discussed in the context of large landslide occurrences: Marc et al., 2019 – Esurf

I am looking forward to seeing this manuscript published in a revised form. Maarten Lupker

---

## Author Comment (AC1) · 4 Jun 2019

REFEREE: The paper by Lujendra Ojha, Ken L. Ferrier and Tank Ojha presents seven 10Be-derived estimates of catchment-scale denudation rates in Western Nepal. These denudation rates fall within the range of values published elsewhere in Himalaya. The paper is clear, well written and illustrated.

It seems that the main interest of this contribution is to provide new denudation data in a region where they were lacking. The authors do not claim to revolution the debate between the climatic and tectonic control of denudation with their data, which is fair, but the scientific justification of the study is a bit short. Why was it "important" to fill this gap of data? Is it a key place? Was the sampling initially designed for some particular

question? The last sentence of the conclusion says that this study "illustrates the need for future denudation rate measurements in the region to test hypotheses". As it, I am not sure that this is the case. I agree that new data are always useful and that is why I think that this paper should be published. Nevertheless, this study shows that these new denudation data are not able to discuss the relative contribution between climate and tectonics. Thus where do we need to gather future samples to answer this question and why?

RESPONSE: We appreciate the reviewer's suggestions on how to strengthen the motivation for this study, particularly in describing why the study area is of interest and where future sampling should be focused. To address this, we added two paragraphs to the text, one at the end of the Introduction, which describes why this study area is of interest, and one at the end of Section 7, which describes where future sampling efforts may be most fruitfully directed. We also added a new figure (now Figure 2), as suggested by another reviewer, that shows where our samples are located within profiles of topography and stream power across this section of the Himalaya. The new text at the end of the Introduction is as follows.

"Previous studies suggest that the relative strengths of the controls on denudation rate in Far Western Nepal may differ from those in central Nepal. In central Nepal, the presence of a single, major mid-crustal ramp in the Main Himalayan Thrust (MHT) (e.g., Schulte-Pelkum et al., 2005; Bollinger et al., 2006; Nábělek et al., 2009; Elliott et al., 2016) has given rise to a steep topographic gradient with spatially focused exhumation and orographic precipitation (van der Beek et al., 2016). In Far Western Nepal, by contrast, the topography rises more gradually and induces a less intense focusing of orographic precipitation, and has been hypothesized to be a reflection of two distinct mid-crustal ramps, each smaller than the one in central Nepal (Harvey et al., 2015; van der Beek et al., 2016). This is consistent with apatite fission-track thermochronometric measurements that show that Myr-scale exhumation rates and specific stream power are significantly higher and more spatially focused in central Nepal than in Far

Western Nepal (van der Beek et al., 2016). To the extent that along-strike variations in uplift and orographic precipitation influence the spatial patterns and magnitudes of denudation rates, they may also induce along-strike variations in the feedbacks between climate, tectonics, and topography. In this study, we report new basin-averaged denudation rate measurements inferred from cosmogenic 10Be in stream sediment in Far Western Nepal to better understand denudation rate patterns in this segment of the Himalaya. Our measurements show that denudation rates in these basins are consistent with those both east and west of Far Western Nepal, suggesting similar controls on denudation across this portion of the Himalayan arc over millennial timescales, and they highlight the regions that may be most useful to target for future denudation rate measurements."

The following is the new text that has been added to the end of Section 7, which describes where it would be most useful to collect future samples.

"While our measurements are unable to isolate the controls on denudation rates on their own, they are nonetheless able to highlight regions in Far Western Nepal that may be useful targets for future denudation rate measurements. For example, the regions northeast of 100 km and southwest of 180 km in the swath in Figure 2 have few denudation rate measurements, and targeting small basins within these regions may be particularly instructive. For example, to isolate the sensitivity of denudation rates to rock uplift rate, it may be useful to target basins like Raduwa in the Subhimalaya and the southwestern portion of the Lesser Himalaya (Figure 3A), where topographic relief and rock uplift rates are hypothesized to be low, to determine whether the low denudation rate in Raduwa is typical of other basins undergoing similar tectonic forcing. Similarly, it may be helpful to target small catchments that lie over the hypothesized mid-crustal ramps (Harvey et al., 2015), where rock uplift rates may be higher, including small basins at relatively high elevation in the Greater Himalaya within the Karnali, Bheri, and Seti River basins (Figure 3B). Likewise, to assess the sensitivity of denudation rates to stream power, it may be useful to target small basins that isolate regions of high

SSP, like those at mid-elevations in the Bheri catchment and high elevations in the Seti catchment, as well as basins that isolate regions of low SSP, like those at low elevation in the southwestern portions of the Bheri, Karnali, and Seti basins (Figure 4C). While some of these sites are relatively remote and were beyond our ability to access them during this study, future field campaigns that are able to collect samples from these sites may be able to put stricter constraints on the couplings between denudation rate, rock uplift, and climate in Far Western Nepal."

REFEREE: The calculation of the denudation rates is correct and includes a good discussion of uncertainties. Yet, some aspects of these uncertainties could be improved: Dibiase (Earth Surf. Dynam., 6, 923-931, 2018) recently showed that the topographic shielding correction is usually unappropriated. As denudation values with and without topographic shielding are already given, the authors could only recall the Dibiase's paper.

RESPONSE: We agree that an accurate assessment of topographic shielding can be significant, especially in exceptionally steep topography, as DiBiase (2018) showed. To the extent that the model geometry adopted by DiBiase (2018) applies to our study basins, where our estimates of topographic shielding are relatively small (ranging from 0.6% to 2.5% among basins), this would increase our estimates of denudation rate by < 2.5%. To address this, we added the following text at Line 50 of Section 3.2.2.

"Recently, DiBiase (2018) showed that this approach can overestimate the extent of topographic shielding, particularly in steeply dipping catchments, and argued that topographic shielding factors should be 1 in basins with horizontal surrounding ridges. If this horizontal ridge geometry is applicable to our study basins, where our estimates of topographic shielding range from 0.9759 to 0.9939 (Table 2), then the denudation rates in Table 2 would be underestimated by 0.6% to 2.5%."

REFEREE: The uncertainty on production rate is not indicated. It can easily reach more than 10% and depends on the production rate model (which one was used in the

CRONUS calculator?) and thus should be propagated to the denudation rate uncertainty.

RESPONSE: Uncertainties in the production rate are incorporated directly into the denudation rate estimates computed by the CRONUS calculator, which it reports as the "external" uncertainty, and which does not report the production rate uncertainty separately. To address this, we added the following text at Line 40 in Section 3.2.3.

"The denudation rate uncertainties in Table 2 are the "external" uncertainties reported by the CRONUS calculator for the Lal/Stone production rate scaling scheme, and include propagated uncertainties on the 10Be production rate."

REFEREE: What is exactly the maximum grain size of quartz that was dissolved? Table S1 gives the distribution of grain sizes for each sample. I understand that all these grain sizes were dissolved together. Why doing this, while many studies have shown that the 10Be concentration is grain size dependent? The grain size distribution differs a lot between catchments. Does denudation rate correlate with the mean (or other metrics) grain size in this dataset, as observed in many other cases?

RESPONSE: As noted in Table S2, most of the samples were dominated by sand-sized sediment. We did not measure the size of the largest grains, but we estimate that the largest grains at Raduwa (which has the largest median grain size among our samples) were no larger than 40-50 mm in diameter. We analyzed all grain sizes in the proportion they were present in our samples for precisely the reason the reviewer notes, i.e., that 10Be concentrations are often inversely related to grain size (e.g., Brown et al., 1995, EPSL, p. 193-202). Analyzing only a single grain size would yield a biased estimate of the basin-averaged 10Be concentration and hence a biased estimate of denudation rate (e.g., Riebe et al., 2015, PNAS, p. 15,574-15,579). To avoid introducing this bias, we analyzed all grain sizes in our samples to obtain a representative estimate of the mean 10Be concentration. To clarify this, we added the following text at Line 44 in Section 3.1.

"We analyzed quartz in all sediment grain sizes to avoid introducing biases that would be associated with analyzing only a single grain size (e.g., Brown et al., 1995; Riebe et al., 2015)."

REFEREE: The lithological effect is nicely discussed by exploring two end-members scenarios where only the catchment head or the catchment foot provides quartz. However, how do the relationships between denudation and slope, steepness and stream power change when restricted to the upper or lower catchment parts? For example, by restricting the calculation of ks and mean stream power to the upper Budhiganga catchment, its "anomalous" high denudation may be shifted to the right on diagrams of Figure 4, in a more "classical" configuration. In the studied catchments there is a correlation between lithology (possibly quartz content) and elevation (10Be production rate), as elsewhere along the range. Could it be possible that the lithological effect explains the large dispersion observed between denudation and steepness (see for example Carretier et al., Earth Surface Processes and Landform, 2015)?

RESPONSE: We agree that lithology varies within each catchment, but we do not have sufficient information on quartz content in each lithologic unit to assess how much this might affect the denudation rate estimates, so we refrain from speculating on that issue here. As the manuscript notes at Line 30 in Section 5.4, "We are unaware of published quartz abundances in the underlying lithologies, so we do not recalculate denudation rate estimates for our study basins here." To illustrate how this might affect the estimates of basin-averaged ksn and specific stream power and hence the patterns in Figure 4, we expand on the hypothetical scenario for the Kalanga basin by adding the following text at Line 48 in Section 5.4.

"In this scenario, the inferred ksn and specific stream power estimates would be 46% and 44% higher in the high-elevation portion of the Kalanga basin, respectively, and 33% and 39% lower in the low-elevation portion of the Kalanga basin, respectively. Like the differences in the inferred denudation rates, these differences are relatively small compared to the scatter in the trends in Figure 5, suggesting that these effects

**ESurfD**
are unlikely to be a major source of scatter in Figure 5."

REFEREE: The discussion on the erodibility of different rocks in the Budhiganga and Kalanga catchments, ruling out lithology as possible control of their different denudations, is maybe a bit short (paragraph 20). Are the crystalline rocks of the Budhiganga catchment weathered or fractured? Are the carbonates of the Kalanga layered and possibly more easy to erode? What do other studies in the region or close say about the lithological control of denudation (e.g. Lave and Avouac, JGR, 2001)? Furthermore, this discussion may change by restricting the calculation of ks and stream power to the quartz-rich lithology (previous comment).

RESPONSE: We agree that more detailed characterization of the erodibility of the local rock units would be useful for assessing lithologic effects on denudation rate in the study region, but we are unaware of measurements of the erodibility of these units that would be necessary to assess this. To address this, we expanded this section so that it now reads as follows at Line 18 in Section 7.

"We are unaware of measurements of the erodibility of each lithology in these basins, but current geologic maps suggest the difference between the inferred Budhiganga and Kalanga denudation rates is unlikely to reflect a lithologic control. The Budhiganga basin is dominated by crystalline rocks (migmatitic and calc-silicate gneiss, metavolcanics, orthogneiss, schist, and granite; Figure S1; Gansser, 1964; Arita et al., 1984; Upreti and Le Fort, 1999; DeCelles et al., 2001; Robinson et al., 2006), which tend to be relatively strong and inhibit rapid erosion, while the Kalanga basin is dominated by carbonates, which tend to be less resistant to erosion (Sklar and Dietrich, 2001). Thus, if denudation rates were dominantly controlled by lithology, then to the extent that the mapped lithologic units have erodibilities similar to those of analogous rocks in laboratory tests (e.g., Sklar and Dietrich, 2001), denudation in the Kalanga basin should be faster than that in the Budhiganga, not slower. Evaluating lithologic effects on denudation rate in the study area more rigorously will require new detailed geologic mapping and new measurements of lithologic characteristics (e.g., tensile strength,

fracture spacing, bedding thickness) in these units."

REFEREE: I feel that the answers to these comments are quite straightforward. I encourage the authors to add the suggested analysis.

---

## Author Comment (AC2) · 4 Jun 2019

REFEREE: This article by Ojha and co-authors presents a new dataset of basin-wide denudation rates derived from 10Be measurements in rivers sands. The dataset consists in 7 new 10Be concentration measurements for the Far Western Nepal region in the Himalayas. While numerous studies have reported CRN derived denudation rates in various parts of the Himalayan arc, this is one of the first dataset from this particular area. The data and the methods are well presented and the authors discuss thoroughly the various hypothesis and caveats when calculating the denudation rates from the 10Be concentration measurements. They compare their results with available data in other areas along the Himalayan arc, and then discuss the relative contributions of various types of forcings to denudation rates. Seven samples is quite a small dataset

when compared with other similar studies in this area. However, this is a very important and understudied part of the arc (as noted by the authors this area is difficult to access). Indeed, the region illustrates the existence of important along-strike variations, in particular with respect to the intensively documented central Nepal sections. Any new data is thus a very welcome addition to the body of knowledge of the dynamics of the Himalayas, and has the potential to provide critical constraints on future discussion of the lateral variability along the arc.

My main concern is that, in its present form, the article lacks the formulation of a clear problem statement. The abstract and in particular the introduction read like the primary focus of the article is just to present a new dataset (see for example line 8), which is not a very attractive prospect for potential readers. For example, recent studies highlight a number of peculiar features of Far West Nepal (Harvey et al., 2015, van der Beek et al., 2016) and it might provide a starting point to present the results in terms of the analysis of lateral variations along the arc, in particular with respect to the much better constrained central Nepal area. Concerning the discussion of the implications on the effective controls on denudation, I feel that figure 4 only provides a very blurred and generalized perspective on the problem, and is not a very robust support for this discussion at the scale considered here. Comparing cross sections for denudation data, topography, geology and precipitation (etc . . .) in Far West and Central Nepal, could be very helpful for that purpose. See for example figure 2 of van der Beek et al. (2016).

Specific comments keyed to line numbers

p1-l38 : at this stage of the introduction you should highlight clearly why this region is important and interesting.

p2-l1-8 : the few studies that have looked in details at this area (van der Beek et al., 2016 and Harvey et al., 2015) have articulated a clear problem statement, and you could build on that to reformulate your introduction.

**ESurfD**
RESPONSE: Thank you for these constructive suggestions. To address this, we added the following text at the end of the Introduction, and we added a new figure (now Figure 2) that shows profiles of topography and specific stream power perpendicular to the range across Far Western Nepal.

"Previous studies suggest that the relative strengths of the controls on denudation rate in Far Western Nepal may differ from those in central Nepal. In central Nepal, the presence of a single, major mid-crustal ramp in the Main Himalayan Thrust (MHT) (e.g., Schulte-Pelkum et al., 2005; Bollinger et al., 2006; Nábělek et al., 2009; Elliott et al., 2016) has given rise to a steep topographic gradient with spatially focused exhumation and orographic precipitation (van der Beek et al., 2016). In Far Western Nepal, by contrast, the topography rises more gradually and induces a less intense focusing of orographic precipitation, and has been hypothesized to be a reflection of two distinct mid-crustal ramps, each smaller than the one in central Nepal (Harvey et al., 2015; van der Beek et al., 2016). This is consistent with apatite fission-track thermochronometric measurements that show that Myr-scale exhumation rates and specific stream power are significantly higher and more spatially focused in central Nepal than in Far Western Nepal (van der Beek et al., 2016). To the extent that along-strike variations in uplift and orographic precipitation influence the spatial patterns and magnitudes of denudation rates, they may also induce along-strike variations in the feedbacks between climate, tectonics, and topography. In this study, we report new basin-averaged denudation rate measurements inferred from cosmogenic 10Be in stream sediment in Far Western Nepal to better understand denudation rate patterns in this segment of the Himalaya. Our measurements show that denudation rates in these basins are consistent with those both east and west of Far Western Nepal, suggesting similar controls on denudation across this portion of the Himalayan arc over millennial timescales, and they highlight the regions that may be most useful to target for future denudation rate measurements."

REFEREE: p2- l30-33 : the presentation of the ksn belongs to the Methods section.

RESPONSE: We respectfully disagree. Section 2's description of the study area includes reference to the steepness index, which warrants definition of ksn here.

REFEREE: section 2 Provide some information about the implication of these lithological variations between the different units in terms of relative quartz abundance.

RESPONSE: As the manuscript discusses this issue in detail in Section 5.4, we do not add further discussion of it here.

REFEREE: p3 - l24 what is the upper limit for the grain size analyzed?

RESPONSE: (Here we repeat our response to a similar comment by Reviewer 1.) As noted in Table S2, most of the samples were dominated by sand-sized sediment. We did not measure the size of the largest grains, but we estimate that the largest grains at Raduwa (which has the largest median grain size among our samples) were no larger than 40-50 mm in diameter. We analyzed all grain sizes in the proportion they were present in our samples for precisely the reason the reviewer notes, i.e., that 10Be concentrations are often inversely related to grain size (e.g., Brown et al., 1995, EPSL, p. 193-202). Analyzing only a single grain size would yield a biased estimate of the basin-averaged 10Be concentration and hence a biased estimate of denudation rate (e.g., Riebe et al., 2015, PNAS, p. 15,574-15,579). To avoid introducing this bias, we analyzed all grain sizes in our samples to obtain a representative estimate of the mean 10Be concentration. To clarify this, we added the following text at Line 44 in Section 3.1.

"We analyzed quartz in all sediment grain sizes to avoid introducing biases that would be associated with analyzing only a single grain size (e.g., Brown et al., 1995; Riebe et al., 2015)."

REFEREE: p3- l35 : this is discussed later, but you should clearly state the fact that you do not take into account the variations in quartz content as a potential limitation.

RESPONSE: To clarify this, we added the following text at Line 18 in Section 3.2.

"In the Discussion section, we describe how our denudation rate estimates may be affected by uncertainties in a variety of factors, including lithologic variations in quartz abundance, which are not well quantified across our study basins."

REFEREE: P5 l20-23 : sentence not clear, "why most applicable?"

RESPONSE: To clarify this, we revised the parenthetical statement in this sentence so that it reads "... (and hence most appropriate for the ïĄč methodology, which was developed for bedrock-dominated channels subject to the stream power law)".

REFEREE: P5 l40-45 : this is interesting and not frequent in this kind of studies, so it might be worth giving a bit more visibility to the corresponding results.

RESPONSE: Thank you for the positive feedback. As the text states, this is part of a separate study in prep that will discuss these results in more detail, so we have left this section as is.

REFEREE: section 5.1 : in the absence of indications on the grain size used here it is difficult to discuss the impact of eventual landsliding contribution on the measured concentration.

RESPONSE: (Here we repeat our response to a similar comment by Referee 3.) We agree that grain size distributions in fluvial sediment are only a coarse reflection of landslide-derived inputs, given the partial filtering of grain size accomplished by fluvial transport. Although we maintain that grain size distributions can partly reflect landslide inputs to fluvial sediment and therefore can provide a useful clue about recent landsliding (e.g., West et al., 2014, EPSL, p. 143-153), we agree that the grain size distributions in our samples are not a strong test of upstream landsliding. We have therefore removed mention of the grain size distributions from this sentence.

REFEREE: P7 l5-20 I would be surprised if the contribution of chemical weathering were significant in this tectonic context and at this scale. I think there are estimates of solutes fluxes in one of the Lupker et al. paper.

RESPONSE: (Here we repeat our response to a similar comment by Referee 3.) We agree that the effects of chemical erosion are likely to be small at these sites. We added the following text at Line 6 in Section 5.2 to address this.

"Similarly, modern fluvial sediment and solute fluxes elsewhere in the Himalaya suggest that the chemical weathering flux in the Ganges and Brahmaputra Rivers is $\sim 9 \pm 2\%$ of the suspended sediment flux (Galy and France-Lanord, 2001) and that chemical weathering fluxes in Himalayan basins may be small relative to those generated in the lowland floodplains (West et al., 2002; Lupker et al., 2012). To the extent that these measurements are applicable to our study basins, this suggests that the chemical erosion may have only a small effect on our denudation rate estimates."

REFEREE: P8 l25 "larger tan 250 microns" is not a very precise definition of the grain size and does not allow to make a robust claim on this point.

RESPONSE: We agree that the influence of grain size on 10Be concentrations can't be strongly constrained from a comparison between our measurements and those of Lupker et al. (2012). This is consistent with one of the intended points of this paragraph, which is that our data do not require that the observed difference in 10Be concentration be attributable to grain size differences, but rather that they are consistent with that possibility. To emphasize that it's possible that grain size differences may be responsible for some of the difference in 10Be concentrations, we revised the sentence at Line 18 in Section 5.5 as follows.

"In addition, it is possible that grain size differences between samples may have contributed to the difference in 10Be concentrations, as Lupker et al. (2012) analyzed quartz grains in the 125-250 micron size fraction, whereas we analyzed only grains larger than 250 microns."

REFEREE: section 6 - even if you have less data it might be interesting, in terms of comparison, to plot you denudation rates along a cross section perpendicular to the range (as well as topography), with similar sections for central and/or eastern Nepal.

section 7 - same as previous points, having cross sections with denudation, topography, precipitation (+ thermochron data, etc. . .), might help to put everything into context, and make the argument easier.

RESPONSE: To address these two related suggestions, we added a new figure (now Figure 2) that shows the locations of our samples within profiles of topography and specific stream power perpendicular to the range across a swath of the Himalayas in Far Western Nepal. Our goal in adding this figure is to provide further context for the study area.

REFEREE: Figure 3B&C : at this scale there might be some overlaps with the data points representing ksn or Spw on individual stream segments, it would probably be better to use a continuous representation (topotoolbox has some function for that purpose). Same comment for S2 and S4.

RESPONSE: We are not sure what the reviewer means by "continuous representation" or what the reviewer is suggesting as an alternative. Regardless, we feel that the figures display the relevant information clearly enough, so we have left them as they are.

REFEREE: Figure 4 : The trends and relationships discussed in the text should be plotted on the figure with their confidence intervals. Figure 2b of Scherler et al (2014) display less scatter than what you plot here, did you subset the data according to some criteria (glaciated, lithology, . . ..)? You could display the trends identified by Olen et al. in adjacent regions. A and B are actually not indicated on the figure.

RESPONSE: We intentionally refrained from including regression lines from this figure, since we are not trying to suggest that the data follow a particular functional form. Instead, our aim is merely to show that denudation rates are broadly positively correlated with both ksn and specific stream power, as described in the text. The gray dots in panel B include data from the compilation of Scherler et al. (2017) and the dataset of Adams et al. (2016), the latter of which had been missing from the figure caption. We

updated the last sentence of the figure caption so that it now reads as follows.

". . . (Scherler et al. (2017) and Adams et al. (2016) in panel A, and Olen et al. (2016) in panel B)."

We added the labels "A" and "B" to the panels, as suggested.

---

## Author Comment (AC3) · 4 Jun 2019

REFEREE: This manuscript adds new cosmogenic 10Be data to the growing number of studies mapping out modern denudation rates across the Himalayan arc and fills a data-gap in Western-Nepal where no such data was published so-far. Calculated denudation rates are at par with other central-Himalayan catchments and the authors discuss it in the context of tectonic and climatic driving processes.

This data-set is relatively modest in size (7 samples) and the study is not very ambitious in the way it is set up. However, it provides new and useful data that is worth publishing in my opinion. The authors provide a very detailed and careful discussion of the limitation of the cosmogenic nuclide-derived denudation approach, carefully evaluating the different steps which is something I have appreciated. The writing is clear and the figures are informative. I, however, think that the rationale for the study could be improved so that it reads less like a simple data-report (which is not a bad thing per se, but in that case, an e-surf research article might not be the most appropriate format).

As it currently stands, the authors are focusing on the climate vs. tectonics debate to motivate their study. But I feel that with a limited number of additional samples and a relatively short analysis they do not contribute much to this discussion. This would require a more thorough re-analysis of all 10Be along the Himalayan arc than what is presented here. On the other hand, the authors could have focused in more details on the specificities of the Karnali catchment with older AFT ages and lower stream power values compared to other parts of the range (e.g. van der Beek et al., 2016 – Geology). How these characteristics may or may not be expressed in surface catchment denudation rates seems a worthy discussion angle for this manuscript that is not necessarily very well addressed in this submission.

RESPONSE: We appreciate the reviewer's suggestions on how to strengthen the motivation for this study. To address this, we added the following text to the end of the Introduction, which provides more information for why we focus on Far Western Nepal in this study. To provide further context, we also added a new figure (now Figure 2) that shows where our samples are located within profiles of topography and stream power across this section of the Himalaya.

"Previous studies suggest that the relative strengths of the controls on denudation rate in Far Western Nepal may differ from those in central Nepal. In central Nepal, the presence of a single, major mid-crustal ramp in the Main Himalayan Thrust (MHT) (e.g., Schulte-Pelkum et al., 2005; Bollinger et al., 2006; Nábělek et al., 2009; Elliott et al., 2016) has given rise to a steep topographic gradient with spatially focused exhumation and orographic precipitation (van der Beek et al., 2016). In Far Western Nepal, by contrast, the topography rises more gradually and induces a less intense focusing of orographic precipitation, and has been hypothesized to be a reflection of two distinct

mid-crustal ramps, each smaller than the one in central Nepal (Harvey et al., 2015; van der Beek et al., 2016). This is consistent with apatite fission-track thermochronometric measurements that show that Myr-scale exhumation rates and specific stream power are significantly higher and more spatially focused in central Nepal than in Far Western Nepal (van der Beek et al., 2016). To the extent that along-strike variations in uplift and orographic precipitation influence the spatial patterns and magnitudes of denudation rates, they may also induce along-strike variations in the feedbacks between climate, tectonics, and topography. In this study, we report new basin-averaged denudation rate measurements inferred from cosmogenic 10Be in stream sediment in Far Western Nepal to better understand denudation rate patterns in this segment of the Himalaya. Our measurements show that denudation rates in these basins are consistent with those both east and west of Far Western Nepal, suggesting similar controls on denudation across this portion of the Himalayan arc over millennial timescales, and they highlight the regions that may be most useful to target for future denudation rate measurements."

REFEREE: Some more detailed considerations:

- It seems that the authors try to provide a global overview of 10Be data available across the Himalayan range (Figure 1). This is useful but is incomplete. At least 4 papers (maybe more) were omitted and should be mentioned: Puchol et al., 2014 – Geomorphology; West et al., 2015 – Esurf; Lupker et al., 2017 – Esurf; Dingle et al., 2018 – Esurf.

RESPONSE: Thank you for pointing these out these omissions. To address this, we added the locations of these studies to the map in Figure 1. (The sites in West et al. (2015) had in fact already been plotted in Figure 1, but the figure caption had incorrectly cited them as West et al. (2014), so this citation has been changed to West et al. (2015) in the caption.) We also added the relevant citations to the figure caption and the corresponding list of citations in the first sentence of Section 6.

REFEREE: - The comparison between this dataset and published data (e.g. Figure 4) should be made carefully. If I understand it correctly the denudation rate, as well as steepness or stream power, have not been recalculated in a homogeneous way for different datasets. Given that the authors report some large variations between different approaches (e.g. snow cover effect) it is uncertain how these differences may bias this type of comparison. I would suggest to recalculate the data using a homogeneous procedure (even though I am aware that this represents a significant amount of work) or convincingly show that the differences are minor.

RESPONSE: We agree that care needs to be taken in comparisons between denudation rate estimates computed in different studies, since there can be ~10-40% differences in denudation rate estimates computed with different production rate parameters (e.g., Mudd et al., 2016). This is relevant in Figure 5, which aims to compare denudation rate estimates at our study sites in Table 2 (which were computed with CRONUS v2.3) to the denudation rate estimates we compare them to from the literature (Scherler et al. (2017), Adams et al. (2016), and Olen et al. (2016)), all of which were computed with CRONUS v2.2, which used different production rate parameters than those in CRONUS v2.3. As the reviewer notes, recalculating denudation rate estimates for these other studies would be a significant task, particularly because not all of these studies reported each basin's effective elevation or the degree of shielding by ice and seasonal snow, which would be needed to recalculate denudation rates under the same procedure we used. We therefore do not attempt to recalculate denudation rates from those studies here. Instead, for the purpose of making the visual comparison between the datasets in Figure 5 without this source of bias, we recalculated the denudation rate estimates at our study sites using CRONUS v2.2 (the same version that was used in the other studies), and we modified Figure 5 to show these denudation rate estimates alongside the denudation rate estimates in the other studies. The recalculated denudation rate estimates from CRONUS v2.2 are an average of 19% (range: 16-26%) higher than the values computed with CRONUS v2.3 in Table 2. Because these estimates follow the same broad patterns that were visible in these data before

[Figure]

ESurfD
this recalculation, our central interpretations of Figure 5 are unchanged from those in the previous version of this manuscript. To make it clear that all denudation rate estimates in Figure 5 were calculated using the same version of the CRONUS calculator, we added the following text to the end of the figure caption.

"To avoid introducing biases to comparisons of denudation rate estimates determined from different versions of the CRONUS calculator, the black dots in Figure 5 show denudation rate estimates at our study sites that have been recalculated using the same version of the CRONUS calculator (v2.2) as that used in Adams et al. (2016), Olen et al. (2016), and Scherler et al. (2017). These rates are an average of 19% (range: 16-26%) higher than those calculated with CRONUS v2.3 in Table 2."

REFEREE: - The use of a topographic shielding correction in catchment-wide denudation rates has been recently questioned DiBiase, 2018 – Esurf

RESPONSE: (Here we repeat our response to a similar comment by Referee 1.) We agree that an accurate assessment of topographic shielding effect can be important, especially in exceptionally steep topography, as DiBiase (2018) showed. To the extent that the model geometry adopted by DiBiase (2018) applies to our study basins, where our estimates of topographic shielding are relatively small (0.6 to 2.5% among basins), this would increase our estimates of denudation rate by < 2.5%. To address this, we added the following text at Line 50 in section 3.2.2.

"Recently, DiBiase (2018) showed that this approach can overestimate the extent of topographic shielding, particularly in steeply dipping catchments, and argued that topographic shielding factors should be 1 in basins with horizontal surrounding ridges. If this horizontal ridge geometry is applicable to our study basins, where our estimates of topographic shielding range from 0.9759 to 0.9939 (Table 2), then the denudation rates in Table 2 would be underestimated by 0.6% to 2.5%."

REFEREE: - p.6, l.52: it is not clear to me how grain-size data on fluvial sediments will tell you much about the importance of landslide inputs given transport segregation

processes.

RESPONSE: We agree that grain size distributions in fluvial sediment are only a coarse reflection of landslide-derived inputs, given the partial filtering of grain size accomplished by fluvial transport. Although we maintain that grain size distributions can partly reflect landslide inputs to fluvial sediment (e.g., West et al., 2014, EPSL, p. 143-153) and therefore can provide a useful clue about recent landsliding, we agree that the grain size distributions in our samples are not a strong test of the prevalence of upstream landsliding. We have therefore removed mention of the grain size distributions from this sentence.

REFEREE: - On the effect of chemical erosion on 10Be denudation estimate (p.7, section 5.2): the fact that chemical denudation is only a very small fraction of the overall mass export (as mentioned later in the manuscript) should provide a rough estimate on the magnitude of this bias.

RESPONSE: We agree that the effects of chemical erosion are likely to be small at these sites. We added the following text at Line 6 in Section 5.2 to address this.

"Similarly, modern fluvial sediment and solute fluxes elsewhere in the Himalaya suggest that the chemical weathering flux in the Ganges and Brahmaputra Rivers is $\sim 9 \pm 2\%$ of the suspended sediment flux (Galy and France-Lanord, 2001) and that chemical weathering fluxes in Himalayan basins may be small relative to those generated in the lowland floodplains (West et al., 2002; Lupker et al., 2012). To the extent that these measurements are applicable to our study basins, this suggests that chemical erosion may have only a small effect on our denudation rate estimates."

REFEREE: - p.8, l.26: Puchol et al., 2015 – Geomorphology provides a direct example of 10Be concentrations correlated with grain-size induced by landslide processes in a Himalayan catchment.

RESPONSE: We believe this is referring to Puchol et al. (2014), which refers to the

grain-size dependence of 10Be in a Himalayan watershed, rather than Puchol et al. (2015), which we were not able to find a reference to. We added a citation to Puchol et al. (2014) to the list of citations at Line 22 in Section 5.5.

REFEREE: - p.9, l. 1-2 the difference between short-term denudation estimates and long-term rates in the Himalaya has been very recently discussed in the context of large landslide occurrences: Marc et al., 2019 – Esurf

RESPONSE: Thank you for drawing our attention to this recent study from central Nepal. We modified this sentence to include a citation to this study at Line 23 in Section 5.5, which now reads as follows.

"This difference of a factor of 1.5-2 is relatively small compared to the order-of-magnitude differences between short-term and long-term rates often observed in small catchments, particularly those subject to large, rare landslides (e.g., Kirchner et al., 2001; Hewawasam et al., 2003; Covault et al., 2013; Marc et al., 2019)."

REFEREE: I am looking forward to seeing this manuscript published in a revised form. Maarten Lupker

---

## Author Comment (AC6) · 4 Jun 2019

The comment was uploaded in the form of a supplement:
https://www.earth-surf-dynam-discuss.net/esurf-2019-7/esurf-2019-7-AC6-supplement.zip

---

## Author Response (AR2)

Dear Authors,

Your paper was sent back to one referee who is mostly happy with your revision but yet suggests a few minor corrections. Please implement these modifications before we can proceed further.

All the best,

SC

*Dear Editor,*

*Thank you for your detailed attention to this paper. We have made some of the changes suggested by Dr. Maarten Lupker. Please find our detailed response below.*

This is the second time I see this manuscript. I am mostly satisfied with the revisions and responses the authors made to my own comments. I also think that they have made fair and constructive responses to the two other reviewers' comments (many of which were common to the 3 reviews). I, therefore, think that this paper is clear, well written and is in a good state to be published even though the dataset remains of modest size and the study not as ambitious as could have been.

*Thank you for your prompt second review of this paper.*
*We do agree that the dataset remains of modest size. Our hope is that this pilot study offers motivation for future studies to focus in the Himalayas of the Far-Western Nepal.*

I have a few remaining comments:

- I don't think that analyzing all sampled Qz grain-sizes for 10Be is necessarily more representative or better than a given fraction in particular as mentioned in the response to the reviewers. If the aim is to integrate the entire transported sediment load, pebble grain-sizes and above should be considered which remains difficult. A single sample from a bar is also unlikely to be a good representation of the transported load as it will be strongly dependent on local hydrological conditions. On the other hand, a unique grain-size class allows for better comparison between studies since it has been shown on different occasions that 10Be concentrations are grain-size dependent.

*We agree that differences in grain size can complicate comparisons between samples, and that the comparison between our Karnali sample and sample CA10-5 in Lupker et al. (2012) would have been simpler if both samples had the same grain size distribution. We also agree that it is challenging to proportionally sample the entire grain size distribution in a river, an issue that has not disappeared since Brown et al. (1995; Earth and Planetary Science Letters, v. 129, p. 193-202). To the extent that 10Be concentrations vary monotonically with grain size, however, the selective analysis of a single, small grain size is likely to bias the inferred denudation rate more than analyzing a sediment sample that has a grain size distribution closer to the river sediment's true grain size distribution. Since our primary goal is to estimate denudation rate, and because Section*

*5.5 already discusses the potential effects of grain size on cosmogenic nuclide concentrations, we have left the text as is.*

- "We are unaware of measurements of erodibility of each lithology in these basins": Attal and Lavé, 2006 (GSA Special paper 398) provide bedload abrasion rates of various lithologies from very nearby central Nepal...

*Thank you. We were not aware of the above paper.*
*We have revised the manuscript accordingly and have included a reference to the above paper.*

- This has probably a very minor impact on the final values but upon reading the paper again I wondered if the permanent snow cover assumed for elevation above 4920m is justified. I am mainly thinking of the northern part of the catchments on the dry Tibetan plateau.

*Thank you, this is an interesting point. We agree that the parameterization for snow cover is not well constrained by field measurements, as mentioned in Section 5.3, and that this is especially so at high altitude. We acknowledge this point by adding the following sentence to the end of the first paragraph in Section 5.3.*

*"Future snow cover measurements would be particularly useful for constraining the degree of snow shielding at high altitude, especially in the relatively dry Tibetan Plateau, which may have less snow cover than the applied parameterization predicts (Section 3.2.2)."*

Looking forward to seeing the final paper published.

*Thank you again for your detailed review of this paper.*

[revised manuscript text omitted]